# Mutant p53 variants differentially impact replication initiation and activate cGAS-STING to affect immune checkpoint inhibition

Kang Liu[1,2], Lidija A. Wilhelms Garan[1,3], Fang-Tsyr Lin[1,2] & Weei-Chin Lin ✉ [1,2,3,4] ✉

Prior research shows that Akt-dependent phosphorylation of TopBP1 in S phase results in the switch of TopBP1/Treslin binding to TopBP1/E2F1 binding, which is important to prevent replication re-initiation in late S and G2 phases. Here, we demonstrate that contact, but not conformational, mutant p53 can override this switch by binding to both TopBP1 and Treslin, thereby facilitating persistent TopBP1/Treslin interaction in late S and G2 phases, which ultimately leads to over-firing of replication initiation. This increases micronuclei formation, which is further enhanced by genotoxic stressors such as doxorubicin, PARP inhibitors, or ATR inhibitors. Consequently, contact mutant p53 increases the sensitivity of cancer cells to a TopBP1-BRCT7/8 inhibitor combined with PARP or ATR inhibitors. Importantly, contact mutant p53 induces micronuclei formation and MRE11 expression, thereby activating cGAS-STING pathway and enhancing response to immune checkpoint inhibition. This finding is validated in murine mammary tumor allografts and further corroborated by clinical data.

DNA replication is a fundamental process essential for cell proliferation and is tightly regulated during cell cycle progression. The initiation of DNA replication during G1/S transition in mammalian cells is activated by Cdk2/cyclin E, which phosphorylates Treslin and induces its interaction with Topoisomerase IIβ Binding Protein 1 (TopBP1)[1]. The interaction between Treslin and TopBP1 triggers the initiation of DNA replication by uploading Cdc45 onto replication origins[2]. As cells progress through the S and G2 phases, Cdk2/cyclin A activates Akt, which in turn phosphorylates TopBP1 at S1159. This modification shifts the interaction from TopBP1/Treslin to TopBP1/E2F1, thereby terminating replication initiation[3]. The role of TopBP1 in replication initiation is restricted to a specific cell cycle window, with high Cdk2/cyclin E activity but low Akt activity during the G1/S transition and early S phase. Through the sequential actions of Cdk2 and Akt, TopBP1 function is switched from replication initiation to the repression of E2F1-mediated apoptosis during and after mid-S phase.

In addition to terminating replication initiation via the regulation of the TopBP1/Treslin interaction, several other mechanisms have been shown to prevent DNA re-replication. These include downregulation of Cdt1 activity and the displacement of MCM2-7 complexes from replicated DNA[4]. Loss of Cdk activity in the G2 phase can lead to re-replication, highlighting the importance of maintaining moderate Cdk activity after DNA replication to avoid re-initiating the S phase[5]. Since Cdk2/cyclin A and Akt are active between mid-S and G2 phases, the Cdk2/cyclin A-Akt-TopBP1 pathway may play a role in preventing re-replication during G2 phase.

The tumor suppressor p53 protein, referred to as the "guardian of the genome," is central to DNA damage responses. It can induce cell cycle arrest, apoptosis, or senescence, thereby preventing the propagation of damaged DNA. Mutations of *TP53* are found in approximately 50% of human cancers. These mutations lead to the stabilization and accumulation of dysfunctional p53 protein. Unlike the wild-type (WT) p53, mutant p53 (mutp53) loses its tumor-suppressive functions and often gains oncogenic properties, including increased cell proliferation, resistance to cell death, and enhanced metastatic potential. One emerging mechanism by which mutp53 promotes tumor progression involves its interaction with DNA replication machinery, including Treslin[6], MCM proteins[7,8], and Poly(ADP-ribose) polymerase 1 (PARP1)[9], which can disrupt replication dynamics. Through the interaction with MCM proteins, certain mutp53 variants induce replication stress and chromosomal instability, thereby activating the cyclic GMP-AMP synthase (cGAS)-stimulator of interferon genes (STING) pathway[8], a key component of the innate immune response to cytosolic DNA.

[1]Section of Hematology/Oncology, Department of Medicine, Baylor College of Medicine, Houston, TX, USA. [2]Dan L. Duncan Comprehensive Cancer Center, Baylor College of Medicine, Houston, TX, USA. [3]Cancer and Cell Biology Graduate Program, Baylor College of Medicine, Houston, TX, USA. [4]Department of Molecular and Cellular Biology, Baylor College of Medicine, Houston, TX, USA. ✉e-mail: weeichil@bcm.edu

Upon detection of cytosolic double-stranded DNA, cGAS synthesizes cyclic GMP-AMP (cGAMP), which activates STING and triggers downstream signaling cascades that culminate in type I interferon (IFN-I) production and pro-inflammatory cytokine expression. While acute activation of the cGAS-STING pathway supports antiviral defense, tumor surveillance, and responses to genotoxic stress, chronic cGAS-STING activation can lead to endoplasmic reticulum stress, thereby promoting immune evasion and metastasis[10]. Interestingly, although cGAS-STING activation is frequently observed in cancers harboring mutp53, partly due to replication stress and chromosomal instability, this activation ultimately results in pro-tumorigenic inflammation[8]. Conversely, some mutp53 variants can bind TANK-binding kinase 1 (TBK1) to suppress downstream cGAS-STING signaling and facilitate immune escape[11]. An intriguing study comparing $Tp53^{N236S/N236S}$ and $Tp53^{-/-}$ mouse embryonic fibroblasts demonstrates that the N236S variant induces low-level expression of cGAS, IRF9, and IFN-I, which paradoxically impairs the protective cGAS-STING response to acute exogenous DNA exposure, such as during viral infection[12]. These findings underscore the context-dependent effects of mutp53 on cGAS-STING signaling.

Mutp53 can be categorized into contact mutants and conformational mutants based on their structural impact[13]. Contact mutants have alterations that directly affect the DNA-binding surface of the p53 protein, impairing its ability to bind DNA; while conformational mutants cause broader structural changes that disrupt the overall folding and stability of the protein, leading to a loss of normal function. Both contact and conformational mutp53 can bind TopBP1[14] and attenuate the checkpoint response to replication stress[6]. However, only contact mutp53 proteins retain the ability to bind to Treslin. Therefore, during the G1/S transition, contact (but not conformational) mutp53 can bypass the need for Cdk2 to promote replication by facilitating the TopBP1/Treslin interaction[6]. Nevertheless, it remains unknown whether mutp53 can override the inhibitory effect of Akt on TopBP1/Treslin interaction during late S and G2 phases. A prior study demonstrates the in vivo activity of a contact mutp53 in activating cGAS-STING and promoting tumor growth and metastasis[8]. However, it remains unclear how different mutp53 variants differentially modulate the cGAS-STING pathway and whether such modulation may influence responses to immune checkpoint inhibitor therapy.

The interplay between TopBP1 and mutp53 has become a subject of growing interest in DNA replication. The role of TopBP1 in replication and repair processes suggests that its interaction with mutp53 could have significant implications for genomic stability and cancer progression. Understanding these interactions and their effects on DNA replication dynamics is crucial for developing novel therapeutic strategies to target these molecular pathways.

We have now examined the interactions between TopBP1 and either R175H conformational mutp53 or R273H contact mutp53, during S and G2 progression, and their impact on DNA replication. We find that mutp53-R273H, but not mutp53-R175H, facilitates a stable interaction between TopBP1 and Treslin in S and G2 phases. Consequently, mutp53-R273H promotes the initiation of DNA replication in late S phase, resulting in the increased micronuclei formation, an indication of genomic instability. Mutp53 also enhances the expression of MRE11, which can mobilize cGAS from nucleosomes in micronuclei[15]. As a result, mutp53-R273H activates cGAS-STING signaling and enhances the response to anti-PD-1 immune checkpoint inhibitor in a mouse mammary tumor allograft model. Expression of mutp53-R273H in cancer cells also enhances their sensitivity to a combined treatment with a TopBP1-BRCT7/8 inhibitor and PARP or ATR (Ataxia Telangiectasia and Rad3-related protein) inhibitors. These data are clinically significant as contact mutp53, particularly R273H, is linked to increased cGAS-STING activity and improved response to immune checkpoint inhibitors.

## Results

### Contact mutp53 associates with both TopBP1 and Treslin, and facilitates a stable interaction of these two proteins throughout the S and G2 phases

To determine whether mutp53 perturbs the control of TopBP1/Treslin interaction, we first examined the binding of these proteins in the breast cancer cell line MDA-MB-468 that harbors mutp53-R273H. We synchronized cells at the G1/S border by double thymidine block, followed by the release of cells into the S phase for various times as described previously[3]. We then performed co-immunoprecipitation with an anti-TopBP1 antibody (Fig. 1A). We found that TopBP1 interacted with E2F1 at mid to late S phase, coinciding with the phosphorylation of TopBP1 at S1159 residue by Akt. TopBP1 also interacted with mutp53-R273H throughout the cell cycle. Significantly, the interaction between TopBP1 and Treslin persisted throughout the entire S phase (Fig. 1A and Fig. S1A). This result is in sharp contrast with our previous data obtained in wild-type p53-harboring REF52 rat fibroblasts or p53-null H1299 lung cancer cells, where phosphorylation of TopBP1 at S1159 by Akt inhibits TopBP1 from binding to Treslin during the mid- to late S phases of the cell cycle[3]. Thus, phosphorylation of TopBP1 at S1159 failed to prevent TopBP1 from binding to Treslin in the presence of mutp53-R273H. The persistent interaction between TopBP1 and Treslin throughout S phase was seen in another mutp53-R273H-harboring ovarian cancer cell line MDAH-2774 (Fig. 1B and Fig. S1B).

We also examined Treslin/TopBP1 interaction in mutp53-R175H-harboring breast cancer cell line SKBR3 and ovarian cancer cell line TOV-112D (Fig. 1C, D and Fig. S1C, D). TopBP1 interacted with mutp53-R175H in both SKBR3 and TOV-112D cells. However, unlike mutp53-R273H, mutp53-R175H did not perturb the cell cycle regulation of TopBP1/Treslin interaction. The Akt-mediated switch from TopBP1/Treslin interaction to TopBP1/E2F1 interaction during the mid- to late S phases remained intact in both SKBR3 and TOV-112D cells (Fig. 1C, D). SKBR3 exhibited slow growth and had not progressed to mitosis at 10 h after release from the double thymidine block. Consequently, no signal of Ser10 phosphorylation of Histone H3 was detected in Fig. 1C.

It should be noted that in addition to *TP53* mutations, these four cancer cell lines may have differences in other genetic status, which could contribute to the observed differential effect. To further confirm the differential effect of mutp53 variants on the control of TopBP1/Treslin binding, we next expressed either R273H or R175H mutp53 in p53-null H1299 cells. Indeed, the switch remained intact in p53-null H1299 cells and mutp53-R175H-expressing H1299 cells (Fig. 2A, B, Fig. S2A, B). However, the expression of mutp53-R273H rendered persistent interaction between TopBP1 and Treslin throughout S phase (Fig. 2C and Fig. S2C).

Next, we depleted mutp53 in C33A cervical cancer cells that harbor mutp53-R273C, and investigated the effect of mutp53 depletion on the perturbation of this switch. As shown in Fig. 2D, E and Fig. S2D, E, the interaction between TopBP1 and Treslin persisted throughout S phase in C33A cells, but depletion of mutp53 restored the regulation of TopBP1/Treslin binding during S phase progression. Thus, we conclude that mutp53-R273H or R273C, but not R175H, contributes to the constitutive interaction between TopBP1 and Treslin even though TopBP1 is phosphorylated at S1159 by the activated Akt in mid- to late S phase.

Although both mutp53-R175H and R273 (or R273C) interact with TopBP1 throughout S phase (Figs. 1 and 2), our earlier data showed that only mutp53-R273H, but not R175H, can bind Treslin and enhance TopBP1/Treslin interaction to promote DNA replication in G1/S transition even in the presence of a Cdk2 inhibitor[6]. Thus, the differential effect of mutp53 variants on the control of sequential interaction between TopBP1 and either Treslin or E2F1 in mid- to late S phase may also be attributed to the Treslin-binding ability between different p53 mutants. R273H is a contact mutant that loses the DNA-binding activity but preserves the overall p53 structure. R175H is a conformational mutant that may lose part of the tertiary structure required for Treslin binding. To further address the effect of different p53 mutations on the ability to bind Treslin, we also examined

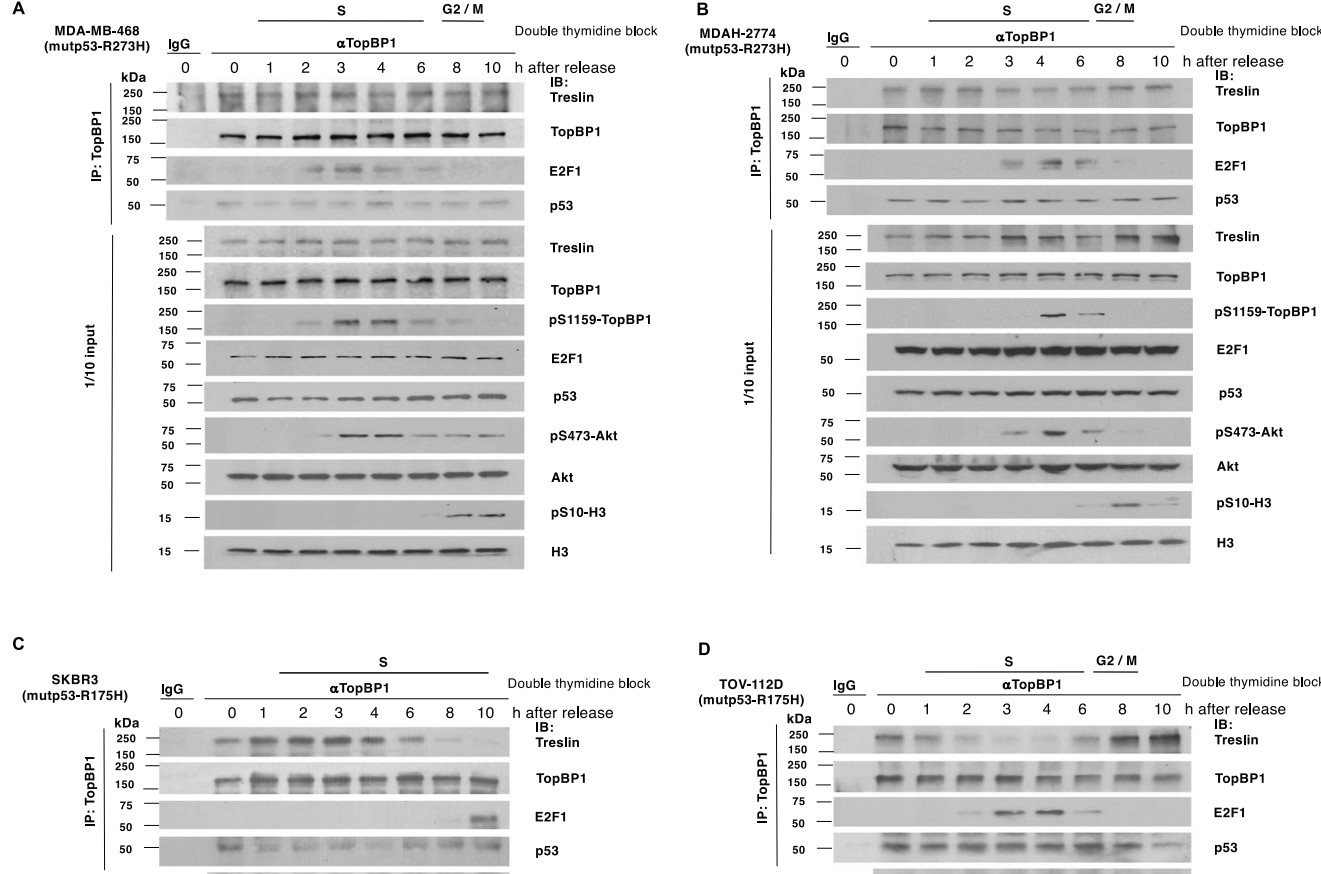

**Fig. 1 | Differential binding of TopBP1/Treslin during the late S phase in cancer cell lines harboring R175H versus R273H mutants of p53.** The breast cancer cell line MDA-MB-468 (**A**) and ovarian cancer cell line MDAH-2774 (**B**), both harboring mutp53-R273H, as well as the breast cancer cell line SKBR3 (**C**) and ovarian cancer cell line TOV-112D (**D**), which harbor mutp53-R175H, were synchronized at the G1/S border using the double thymidine block procedure. After PBS wash twice, cells were cultured in fresh medium containing 10% FBS and harvested in TNN buffer at the designated time points. Co-immunoprecipitation (IP) was performed using an anti-TopBP1 mouse monoclonal antibody or control mouse IgG, followed by immunoblotting. One-tenth of the cell lysates was subjected to Western blot analysis. The bars at the top of each graph represent the samples enriched in S and G2/M phases, respectively. The PI/flow cytometry profiles are shown in Fig. S1. Ser10 phosphorylation of Histone H3 was used as a marker for mitosis.

the interaction of Treslin with another contact mutant, R248W, or conformational mutant, R249S. Indeed, Treslin only interacted with contact mutants R273H and R248W, but not conformational mutants R175H and R249S (Fig. 2F).

**Expression of mutp53-R273H, but not R175H, enhances the initiation of DNA replication during the late S phase**

Next, we performed a double thymidine block procedure to synchronize H1299 cells expressing either an empty vector, mutp53-R175H or mutp53-R273H at the G1/S phase border, followed by the release of cells into S phase (Fig. 3A, B). After 4 h of the release (around late S phase), we labeled newly synthesized DNA with IDU for 30 min, and then with CIDU for another 30 min. Using this sequential labeling protocol, we were able to identify origins of replication fired during labeling[16] (Fig. 3C, upper panel). Our data showed that expression of mutp53-R175H did not change the frequency of replication firing during labeling, but mutp53-R273H significantly promoted replication initiation during the late S phase (Fig. 3C and Fig. S3). We also measured the tract length of the ongoing replication fork, which can be identified as adjacent red and green signals[16] (Fig. 3D, upper panel). It appears that neither R175H nor R273H mutp53 significantly changed the speed of replication (Fig. 3D). In contrast to mutp53-R175H, mutp53-R273H promotes replication initiation, which is in line with its effect on Treslin/TopBP1 binding shown in Figs. 1 and 2.

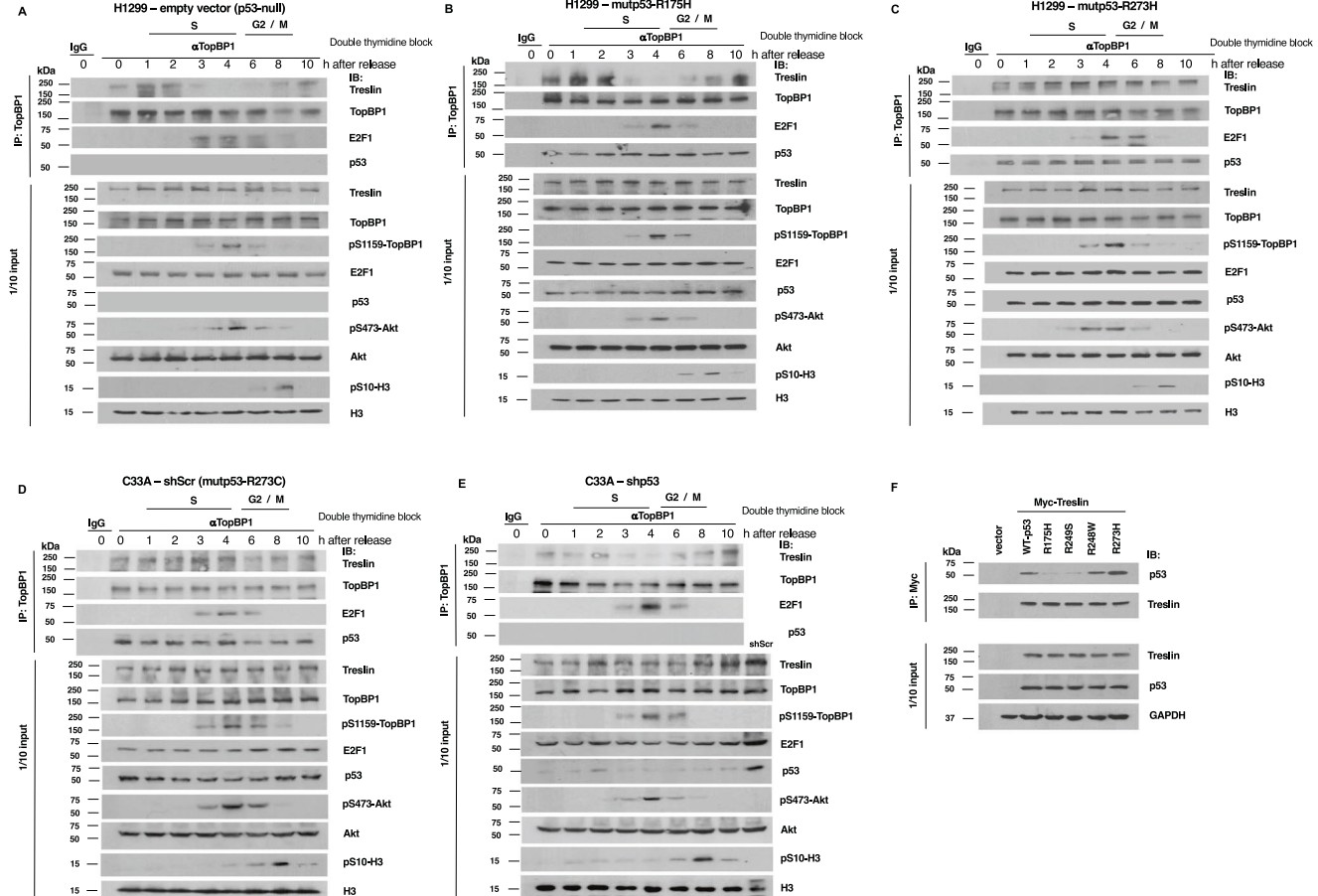

**Fig. 2 | Contact mutp53, but not conformational mutp53, induces the interaction between TopBP1 and Treslin during the S phase of the cell cycle.** H1299 cells expressing a control vector (**A**), mutp53-R175H (**B**), or mutp53-R273H (**C**) were synchronized at the G1/S border using the double thymidine block. Cells were then stimulated with serum for various times, and co-immunoprecipitation was performed with an anti-TopBP1 mouse monoclonal antibody or control mouse IgG, followed by immunoblotting as indicated. One-tenth of the cell lysates was subjected to Western blot analysis. C33A cells (harboring mutp53-R273C) stably expressing either a scrambled shRNA (shScr) (**D**) or mutant p53 shRNA (**E**) were synchronized at the G1/S border using the double thymidine block procedure. After serum stimulation for various times, cells were harvested and subjected to co-immunoprecipitation and Western blot analysis as described above. The PI/flow cytometry profiles are shown in Fig. S2. **F** Myc-Treslin was co-expressed with one of the p53 constructs, including wild-type, conformational mutants (R175H and R249S), and contact mutants (R248W and R273H) in H1299 cells. Co-immunoprecipitation was conducted utilizing Myc-Trap agarose beads. After SDS-PAGE, immunoblotting was performed with the indicated antibodies. One-tenth of the cell lysates was also subjected to Western blot analysis.

## Mutp53-R273H, but not R175H, associates with nascent DNA in S phase

Our prior study showed that mutp53-R273H is localized at the replication forks during DNA replication, supporting a direct involvement of mutp53-R273H in DNA replication[6]. To investigate whether there is a difference between mutp53-R273H and R175H in the association with replication forks, we used Cdk1 inhibitor (Cdk1i) to synchronize H1299 cells harboring either an empty vector, mutp53-R175H or mutp53-R273H at G2/M phase border, followed by the release of cells for 12 h, allowing cells to enter G1/S border as described previously[6]. We then labeled the cells with BrdU or not for 4 h, followed by the anti-BrdU immunoprecipitation from sonicated chromatin. As shown in Fig. 3E, only mutp53-R273H, but not R175H, was pulled down along with PCNA and MCM2 by the anti-BrdU antibody. The complex was associated with nascent DNA, as evidenced by a significant reduction of pulldown following a one-hour BrdU wash-off. This result indicates the association of mutp53-R273H, but not R175H, with actively replicating forks.

## Expression of mutp53-R273H promotes micronuclei formation

The inhibition of Treslin/TopBP1 binding suppresses new replication firing in the late S phase of the cell cycle[3]. The impact of mutp53-R273H on the persistent Treslin/TopBP1 binding throughout the S phase of the cell cycle

may lead to heightened replication firing in the late S phase and cause genomic instability. Thus, we next performed the cytokinesis-block micronucleus (CBMN) assay[17,18], which has been well established for the measurement of genomic instability. Indeed, expression of mutp53-R273H, but not mutp53-R175H, significantly increased the frequency of micronuclei-containing binucleated cells after a short treatment with doxorubicin (Fig. 4). This result provides evidence for perturbing genome stability by mutp53-R273H.

## Mutp53-R273H increases the sensitivity of cancer cells to inhibition of PARP or ATR

The over-firing of DNA replication in late S phase caused by mutp53-R273H might render cancer cells more sensitive to inhibitors that exploit replication fork vulnerabilities. Both PARP inhibitors and ATR inhibitors destabilize replication forks - PARP inhibitors trap PARP1 at DNA damage sites and cause fork stalling and collapse[19], and ATR inhibitors disrupt fork stabilization and promote DNA degradation under replication stress[20]. Therefore, we examined whether mutp53-R273H has any impact on the micronuclei formation induced by PARP inhibitor Talazoparib or ATR inhibitor VE-822. Consistent with the result in Fig. 4, expression of mutp53-R273H in H1299 cells increased the micronuclei formation, and this effect was further enhanced upon treatment with Talazoparib or VE-822 (Fig. 5).

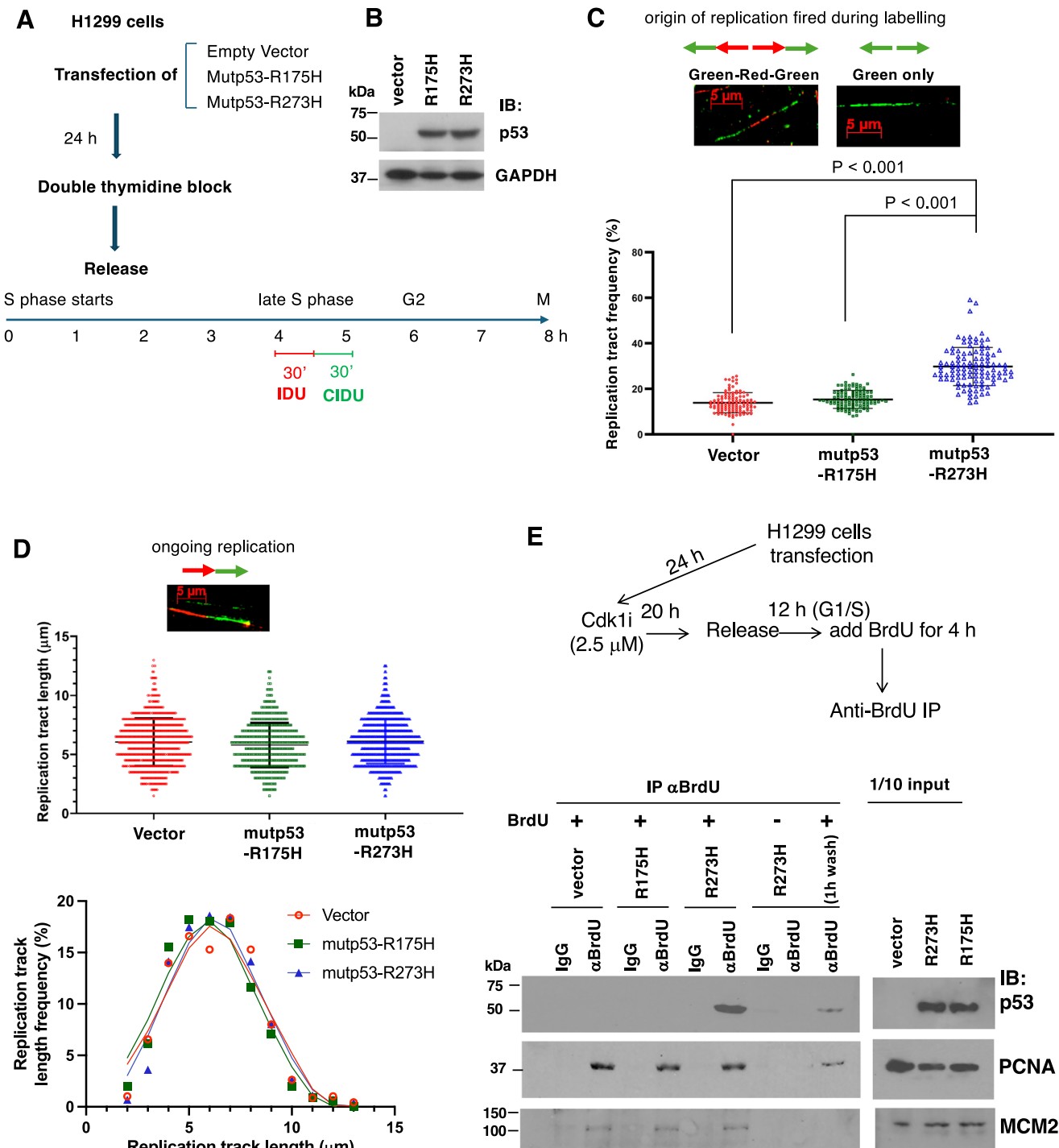

**Fig. 3 | Mutp53-R273H, but not R175H, associates with the replication fork and enhances the initiation of DNA replication during the late S phase.**
**A** Experimental scheme for DNA fiber analysis. **B** Western blot analysis confirms the expression of mutp53 in p53-null H1299 cells. **C** The number of newly formed DNA fibers from origins of replication during the first (green-red-green) and the second (green only) labeling pulses[16] was counted and divided by the total number of ongoing replicated DNA fibers. Over 5000 DNA fibers in more than 100 fields from three biological replicates of each group were analyzed. All p-values are based on a two-tailed *t*-test. Representative images of DNA spreading for (green-red-green) and (green only) are shown in the upper panel and Fig. S3. **D** The length of ongoing replication DNA forks[16] was measured, and the tract length distribution of the vector group, mutp53-R175H group, and mutp53-R273H group was shown. More than 700 DNA fibers were measured for each group. A representative image of DNA spreading for ongoing replication DNA (red-green) is shown in the upper panel. **E** Mutp53-R273H, but not R175H, associates with nascent DNA in S phase. Upper panel: experimental scheme for synchronization of transfected H1299 cells at the G2/M border with a Cdk1i (Ro 3306, 2.5 μM), followed by serum stimulation for 12 h, allowing cells to enter the G1/S phase. After BrdU incorporation for 4 h, cells were harvested, except that one set of cells was washed with PBS and cultured in fresh medium for another hour before harvesting. Anti-BrdU nascent DNA immunoprecipitation was performed, followed by immunoblotting with the indicated antibodies.

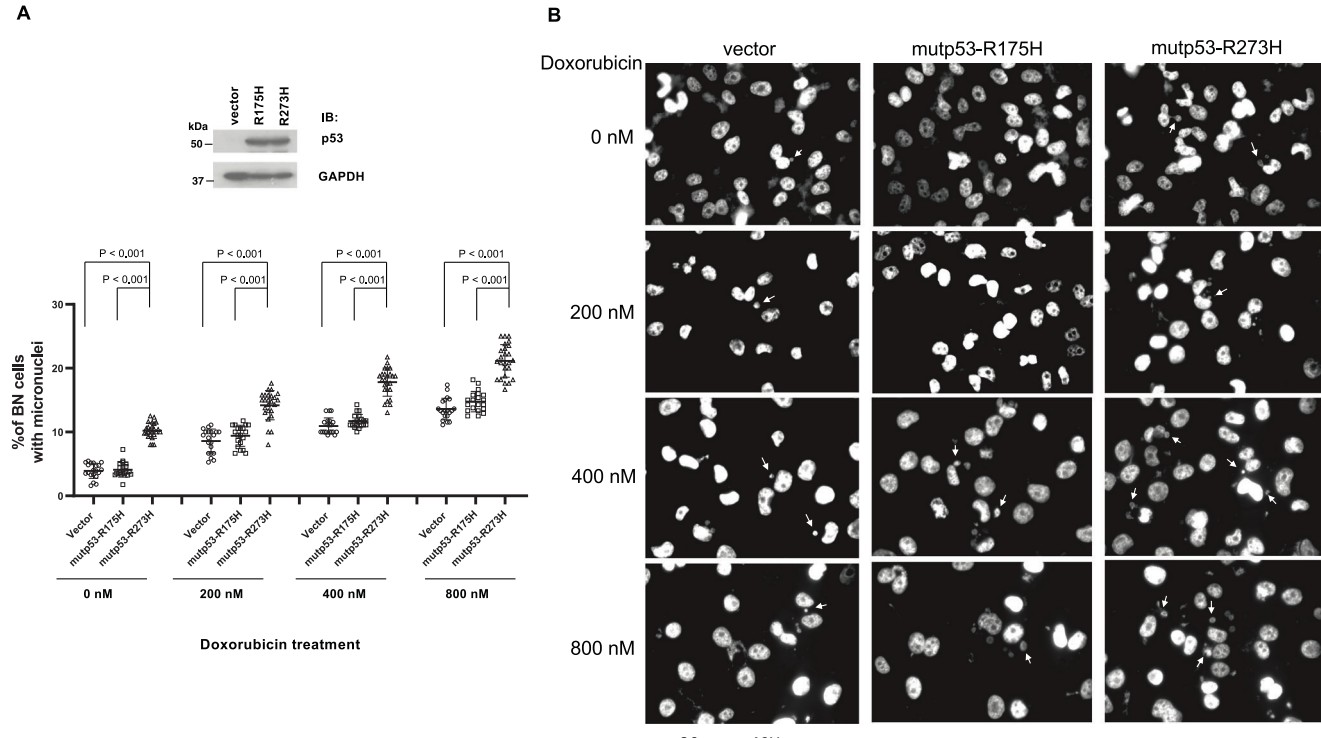

**Fig. 4 | Contact mutp53, but not conformational mutp53, induces micronuclei formation, which is further enhanced by doxorubicin treatment. A** H1299 cell line expressing a control vector, mutp53-R175H, or mutp53-R273H was treated or not with different concentrations of Doxorubicin for 4 h. After the drug was removed, cells were treated with Cytochalasin B (3 μg/ml) for another 24 h, and then fixed with 4% formaldehyde. Nuclei were stained with bisBenzimide, and the number of micronuclei (MN) per binucleated (BN) cell was counted under a microscope. Data represent means and standard deviations from three biological replicates, with p-values calculated using a two-tailed *t*-test. Mutant p53 over-expression was confirmed by immunoblotting. **B** Representative images at 40× magnification for each experimental group. Some micronuclei are labeled with arrows for identification.

Thus, mutp53-R273H may render cells more vulnerable to inhibition of PARP or ATR. We next investigated the effect of mutp53 on the sensitivity to Talazoparib or VE-822. Expression of mutp53-R273H or mutp53-R175H in p53-null H1299 cells only marginally affected the sensitivity to Talazoparib or VE-822 (Fig. 6A, B, left panels). Calcein AM (CalAM), a TopBP1-BRCT7/8 inhibitor, can release E2F1 from the TopBP1 complex and activate E2F1-mediated apoptosis[21]. CalAM can also inhibit the TopBP1/PLK1 interaction and cause a defect in homologous recombination[22], thereby synergizing with PARP inhibitors[23]. CalAM enhanced the sensitivity to Talazoparib and VE-822, and this effect was further promoted by mutp53-R273H, but was mildly decreased by mutp53-R175H (Fig. 6A, B, right panels). Both mutp53-R273H and R175H also differentially affected the induction of DNA damage by Talazoparib or VE-822 without or with CalAM, as reflected by the expression of γ-H2AX (Fig. 6C, D). In the presence of mutp53-R273H, the increased sensitivity to the combination of CalAM with Talazoparib or VE-822 aligns with the increased micronuclei formation due to the perturbation of the replication firing. On the other hand, the effect of mutp53-R175H on the increased drug resistance is consistent with its known "gain-of-function" characteristic.

## Mutp53-R273H activates the cGAS-STING pathway

Micronuclei formation can induce the cGAS-STING pathway[24]. Therefore, we performed qRT-PCR to examine the effect of mutp53 on the induction of the cGAS-STING pathway. Indeed, transient expression of mutp53-R273H in p53-null H1299 cells increased the expression of a number of genes involved in the cGAS-STING pathway, such as cGAS, STING, IFNA1, INFB1, INFG, CCL5, and ISG15, and further enhanced doxorubicin-stimulated cGAS-STING activation (Fig. 7A). Compared to mutp53-R273H, mutp53-R175H only had a very modest effect on the induction of cGAS-STING. Consistently, immunoblotting also showed that

mutp53-R273H, but not mutp53-R175H, promoted the phosphorylation of STING and TBK1 without or with doxorubicin treatment, and increased the levels of p-eIF2α, a marker for immunogenic cell death[25] (Fig. 7B). Despite the induction of *STING* mRNA, STING protein levels did not increase due to strict translational regulation and rapid degradation following activation, preventing excessive immune activation[26]. In contrast, p-STING was induced as the existing STING pool was efficiently activated and phosphorylated before being degraded. The induction of cGAS-STING was also observed in H1299 cells stably expressing mutp53-R273H, but not mutp53-R175H, without or with doxorubicin treatment (Fig. 7C, D). Lastly, we compared the effects of WT-p53, two conformational mutp53 (R175H and R249S), and two contact mutp53 (R248W and R273H) on the induction of cGAS-STING, and found that only contact mutp53 (R248W and R273H) could activate cGAS-STING pathway and elevated the levels of p-eIF2α (Fig. 7E), suggesting a differential regulation of cGAS-STING activation by different mutp53 variants. To further assess the impact of different p53 variants on cGAS-STING pathway activation under isogenic conditions, we first depleted mutp53-R273C expression in C33A cervical carcinoma cells[6] and subsequently reconstituted with either WT, R175H, or R273H variant p53. As shown in Fig. 7F, only the R273H variant, but not the R175H variant, was able to promote cGAS-STING activation, highlighting the variant-specific regulation of this pathway by mutp53. Interestingly, expression of WT-p53 in shScr control C33A cells also elicited a modest activation of cGAS-STING, an effect that is absent in both p53-null H1299 cells (Fig. 7E) and mutp53-depleted C33A cells (Fig. 7F). Previous studies have shown that WT-p53 can facilitate the degradation of cytosolic DNA exonuclease TREX1, leading to the accumulation of cytosolic DNA and subsequent activation of the cGAS-STING pathway[27]. It is plausible that due to mutp53-R273C expression, shScr control C33A cells harbor elevated

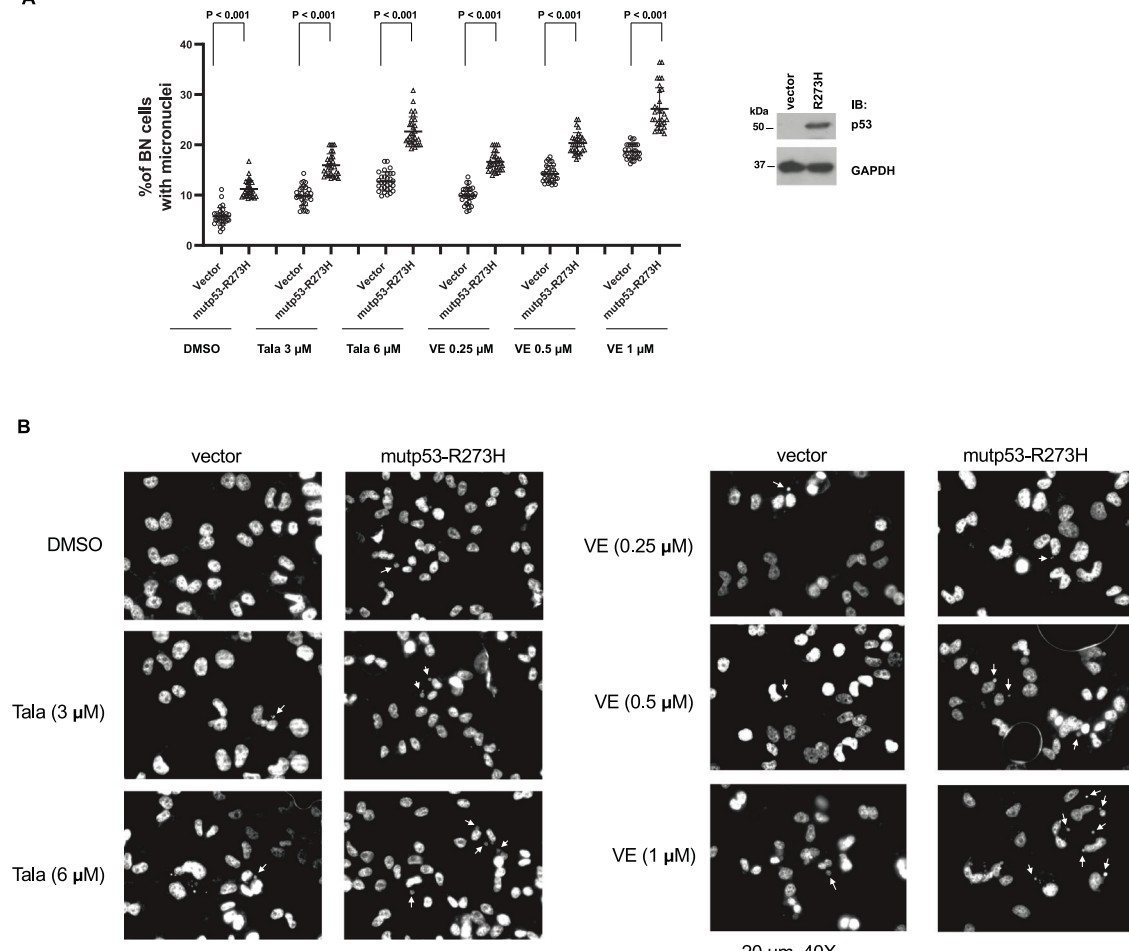

**Fig. 5 | Contact mutp53 promotes micronuclei formation in response to treatment with PARP and ATR inhibitors. A** H1299 cell line expressing a control vector or mutp53-R273H was treated with vehicle control (DMSO) or various concentrations of PARP1 inhibitor Talazoparib (Tala), or ATR inhibitor VE-822 (VE) in the presence of Cytochalasin B (3 μg/ml). After 24 h, cells were fixed with 4% formaldehyde, and nuclei were stained with bisBenzimide. The number of micronuclei (MN) per binucleated (BN) cell was counted under a microscope. Data represent means and standard deviations from three biological replicates, with p-values calculated using a two-tailed t-test. Mutant p53 overexpression was confirmed by immunoblotting. **B** Representative images at 40× magnification for each experimental group. Some micronuclei are marked with arrows.

levels of cytosolic DNA, thereby amplifying the effect of WT-p53 on cGAS-STING activation in this context.

### Mutp53-R273H enhances the response to immune checkpoint inhibitors in 4T1 mammary tumor allografts

The induction of the cGAS-STING pathway and elevation of eIF2α phosphorylation in mutp53-R273H-bearing tumors suggested that these tumors might have a better response to immune checkpoint inhibitors. To test this hypothesis in the immune-competent allograft mouse model, we stably expressed mutp53-R175H or mutp53-R273H in a p53-null mouse mammary tumor cell line 4T1, which is molecularly similar to human triple-negative breast cancer. Three independent stable cell lines for each construct were established. Consistently, the levels of p-TBK1 and p-eIF2α were increased in all three independent 4T1 cell lines harboring mutp53-R273H, but not mutp53-R175H (Fig. 8A).

We next established mutp53-bearing 4T1 orthotopic mammary allografts in BALB/c female mice and treated mice with anti-PD-1 antibody or IgG isotype control twice per week for 3 weeks. Compared to the empty vector controls, the expression of mutp53-R175H or mutp53-R273H led to the development of larger and comparable-sized tumors when treated with control IgG (Fig. 8B–D). Strikingly, mutp53-R273H-bearing 4T1 allografts

showed a better response to anti-PD-1 treatment than the vector group or the mutp53-R175H group (Fig. 8B–D). All mice tolerated the treatment well without body weight loss (Fig. S4A). Compared to the vector controls or mutp53-R175H-bearing 4T1 tumors, the tumor-infiltrating CD8+ T cell populations were significantly increased in mutp53-R273H-bearing 4T1 tumors, especially after anti-PD-1 treatment (Fig. 8E). Immunoblotting also confirmed the induction of p-STING, p-TBK1, p-eIF2α and ISG15 in mutp53-R273H tumors, but not in mutp53-R175H tumors (Fig. 8F). The differential response to anti-PD-1 treatment was not due to a change in PD-L1 expression, since PD-L1 were comparably expressed in 4T1 tumors or H1299 cells harboring mutp53-R273H or mutp53-R175H (Fig. 8F and Fig. S4B).

### Activation of cGAS-STING pathway by mutp53-R273H involves MRE11 and requires TopBP1 and active DNA replication

Although mutp53-R273H can induce micronuclei formation, the formation of micronuclei by itself may not induce the cGAS-STING pathway. Recent findings indicate that although cGAS is recruited to micronuclei following genotoxic stress, its activity is inhibited by nucleosomes[28]. It is now shown that MRE11 binding to nucleosome fragments is crucial for displacing cGAS from acidic-patch-mediated sequestration, allowing its mobilization and

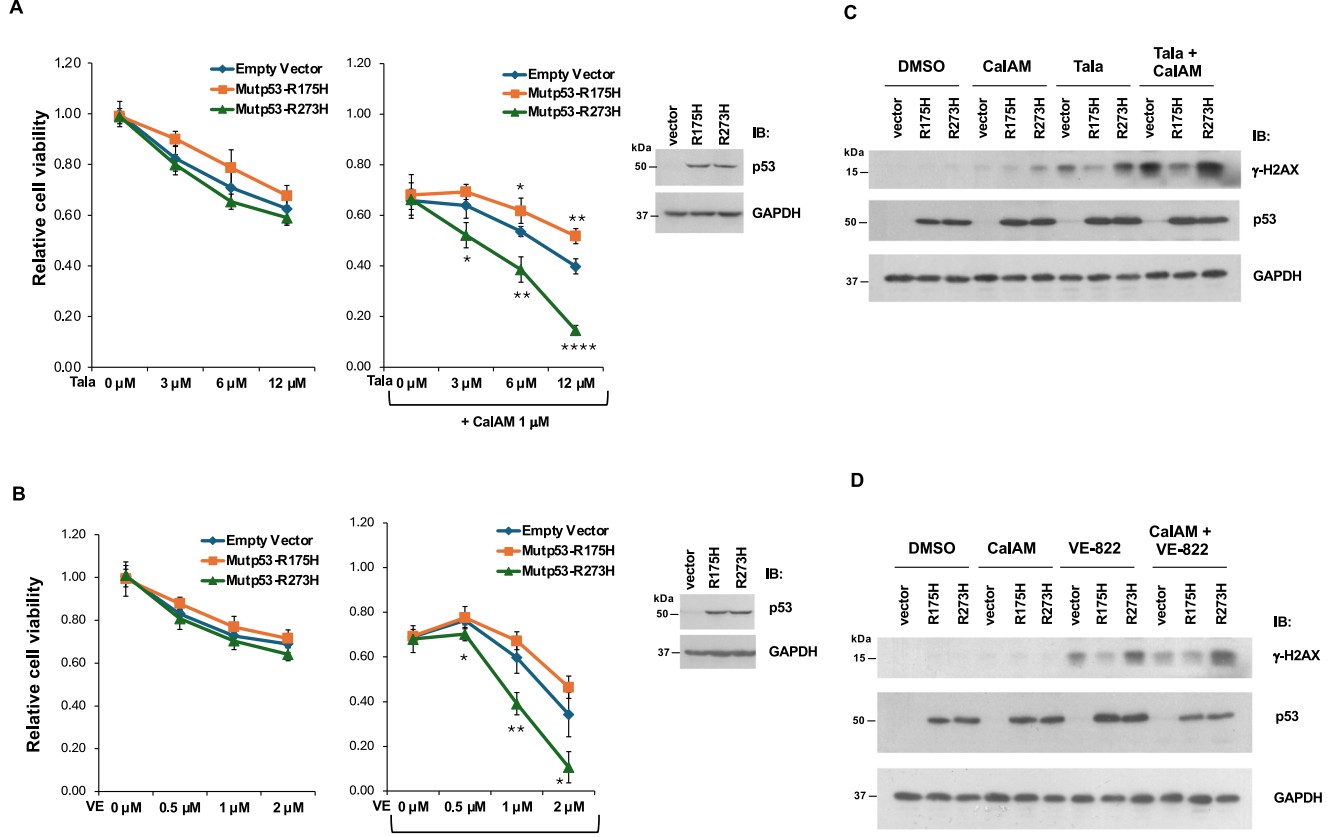

**Fig. 6 | Mutp53-R273H renders cancer cells more sensitive to the combination of Calcein AM with PARP or ATR inhibitors, whereas mutp53-R175H makes them more resistant to this treatment.** H1299 cell line expressing a control vector, mutp53-R175H, or mutp53-R273H was treated with 1 μM Calcein AM (CalAM) and either DMSO or various concentrations of Talazoparib (Tala) (**A**) or VE-822 (VE) (**B**) for 48 h, followed by CCK-8 assay to assess cell viability. Data represent means and standard deviations from three biological replicates, with p-values calculated using a two-tailed *t*-test. *$P < 0.05$, **$P < 0.01$, ***$P < 0.001$, ****$P < 0.0001$, compared with the corresponding control samples. Mutant p53 overexpression was confirmed by immunoblotting. Western blot analysis was performed to determine the levels of γ-H2AX, which serves as a marker for DNA damage response to the treatment of CalAM with either PARP inhibitor (**C**) or ATR inhibitor (**D**).

activation by dsDNA[15]. Based on this finding, we examined public datasets to investigate whether MRE11 is overexpressed in cancer, particularly those harboring mutp53, to potentially liberate cGAS from nucleosomes within micronuclei. Indeed, *MRE11* mRNA and protein are overexpressed in most types of cancer (Fig. S5). It has been reported that MRE11 can be induced by E2F[29,30]. In fact, MRE11, RAD50, and NBS1 are listed among the hallmarks of E2F target genes[31]. We investigated datasets and found additional evidence in support of the notion that deregulation of Rb can upregulate MRE11, RAD50, and NBS1. As shown in Fig. 9A, exogenously expressed E2F1 can induce the expression of MRE11 and RAD50. Consistently, knockout of mouse *Rb1* can upregulate MRE11, RAD50 and NBS1 (Fig. 9B).

We further investigated the relationship between *TP53* mutations and MRE11 expression in human cancer. In most types of cancer, tumors harboring *TP53* mutations express a higher level of MRE11 than the tumors harboring WT-p53 (Fig. S6). The levels of overexpressed MRE11 are similar in tumors harboring either R273H or R175H mutp53 (Fig. 9C). Although MRE11 is not a direct target of p53, its expression can be repressed by WT-p53[32]. Therefore, the loss of WT-p53 may lead to an induction of MRE11. However, the expression of mutp53-R175H or R273H in p53-null H1299 human lung cancer cells resulted in the induction of MRE11 expression (Fig. 9D), suggesting a gain-of-function effect of mutp53 on MRE11 expression. We also examined the MRE11 expression in tumor lysates of 4T1 murine allografts described in Fig. 8. Indeed, both R273H and R175H mutp53 induced MRE11 expression (Fig. 9E). These results indicate that both variants of mutp53 can comparably upregulate MRE11. Based on the promoter occupancy, MRE11 and RAD50, but not NBS1, are listed among

the targets of NF-YA and NF-YB in many ChIP-seq datasets (ENCODE Project Consortium[33]). Consistent with the ChIP-seq data, depletion of NF-Y downregulates the expression of MRE11 and RAD50, but not NBS1 (Fig. 9F), supporting that both MRE11 and RAD50 are NF-Y targets. The transcriptional gain-of-function (GOF) of mutp53 is mediated by the TopBP1/NF-Y/mutp53 complex[14]. TopBP1 broadly enhances the transcriptional GOF of various contact and conformational mutants[14]. To evaluate how TopBP1 contributes to mutp53-R273H-induced cGAS-STING activation, we first depleted TopBP1 in H1299 cells. This depletion effectively abolished the activation of cGAS-STING (Fig. 9G). Similarly, knockdown of TopBP1 in mutp53-R273H-harboring MDA-MB-468 cells led to reduced expression of multiple cGAS-STING target genes (Fig. 9H). To assess the role of DNA replication in this regulation, we then treated cells with a Cdk2 inhibitor (Cdk2i) to halt replication. As shown in Fig. 9I, J, Cdk2i treatment markedly attenuated mutp53-R273H-driven cGAS-STING activation in both 4T1 and H1299 models, confirming that this effect is dependent on active DNA replication.

**Contact mutp53 contributes to the heightened cGAS-STING activation in many types of cancer and is associated with a better response to immune checkpoint inhibitors**

We analyzed the expression of cGAS-STING target genes in the PanCan TCGA database. Compared to the tumors harboring WT-TP53 or mutp53-R175H, the tumors with mutp53-R273H express higher levels of cGAS-STING target genes (Fig. 10A and Fig. S7). The differential expression was also seen in other datasets, including breast cancer METABRIC cohort[34]

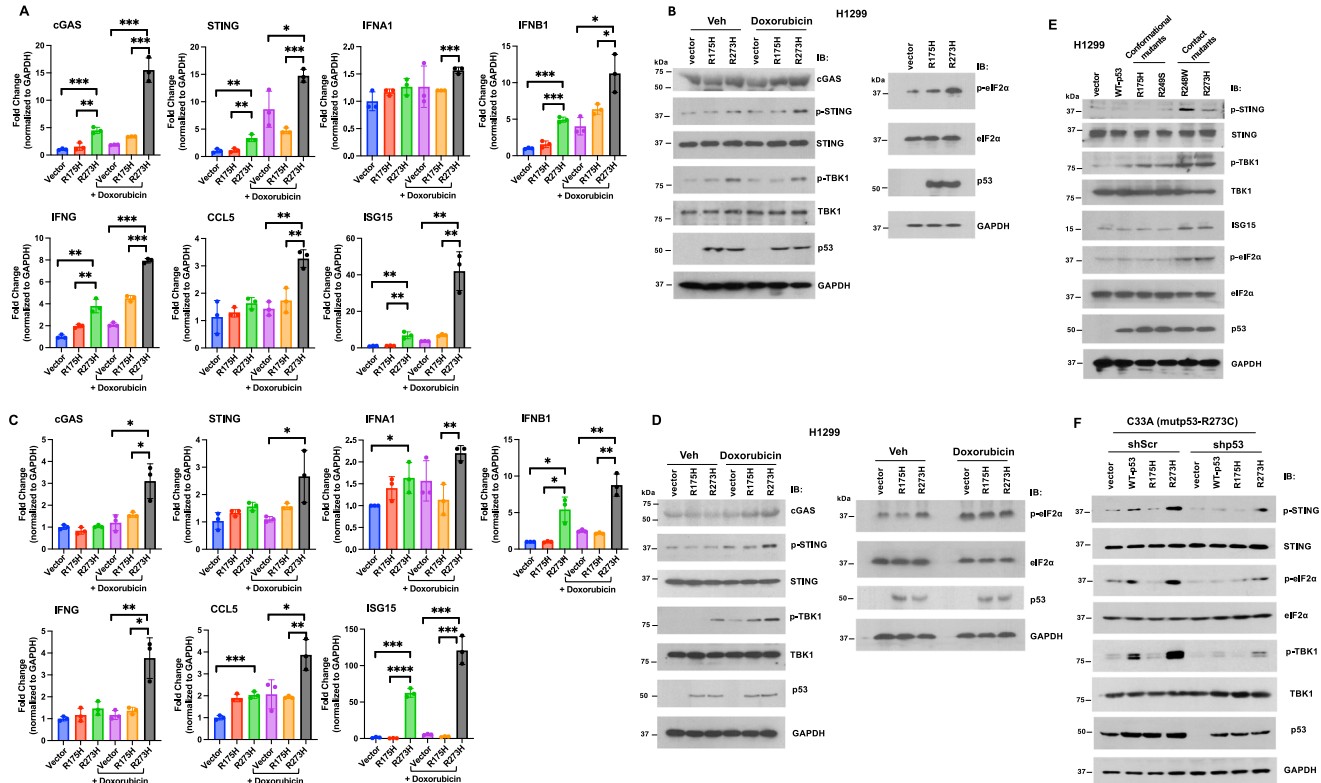

**Fig. 7 | Contact mutp53 increases the expression of genes involved in the cGAS-STING pathway, and further enhances Doxorubicin-stimulated cGAS-STING activation. A** H1299 cell line transiently expressing a control vector, mutp53-R175H, or mutp53-R273H was treated or not with Doxorubicin (800 nM) for 4 h. RNA was extracted, and quantitative real-time RT-PCR was performed using primers specific to *cGAS*, *STING*, *IFN-α1*, *IFN-β1*, *IFN-γ*, *CCL5*, or *ISG15*. The levels of each gene were normalized to *GAPDH* and expressed relative to the vector control. Experiments were performed in triplicate, and data represent means and standard deviations from three biological replicates. p-values were calculated using a two-tailed *t*-test. *$P < 0.05$, **$P < 0.01$, ***$P < 0.001$, ****$P < 0.0001$. **B** The total lysates of H1299 cells, as described in (**A**), were subjected to Western blot analysis to determine the levels of indicated proteins involved in the cGAS-STING pathway. H1299 cell lines stably expressing a control vector, mutp53-R175H or mutp53-R273H were treated or not with Doxorubicin, followed by quantitative real-time RT-PCR (**C**) and Western blot analysis (**D**) as described above to determine the activity of the cGAS-STING pathway in these cells. **E** H1299 cells expressing a control vector, WT-p53, mutp53-R175H, mutp53-R249S, mutp53-R248W, or mutp53-R273H were harvested, and the lysates were subjected to immunoblotting with the indicated antibodies. **F** C33A cells stably expressing either shScr or shp53 were transfected with control vector, WT-p53, mutp53-R175H, or mutp53-R273H. After 48 h, cells were harvested using SDS lysis buffer, and lysates were analyzed by SDS-PAGE followed by immunoblotting with the indicated antibodies.

(Fig. S8), AML[35] (Fig. S9), and metastatic prostate cancer[36] (Fig. S10). In the METABRIC dataset, many cGAS-STING target genes are highly expressed in tumors harboring mutp53-R273H, followed by those harboring mutp53-R175H and WT-*TP53*, respectively (Fig. S8). We then expanded the analysis of the cGAS-STING target gene expression in AML harboring either conformational or contact p53 mutants, and confirmed that AML harboring contact p53 mutants show the highest expression of cGAS-STING target genes compared to those harboring conformational mutp53 or WT-p53 (Fig. S9B). Consistently, analysis of the TCGA CPTAC database also showed higher levels of cGAS-STING-responsive proteins such as CXCL10, OASL, and ISG20 in tumors that harbor mutations in the R273 residue of *TP53* (Fig. 10B).

To investigate the effect of mutp53 on the expression of cGAS-STING target genes, we analyzed two microarray datasets, GSE31812[37] and GSE26262[38], in which mutp53 was depleted in MDA-MB-468 and MDA-MB-231 triple-negative breast cancer (TNBC) cells, respectively. MDA-MB-468 cells harbor mutp53-R273H, while MDA-MB-231 cells carry mutp53-R280K. Both are contact mutants of p53. Indeed, depletion of mutp53 in both cell lines attenuates cGAS-STING target gene expression (Fig. 10C, D). Likewise, depletion of mutp53-R273H in MDAH-2774 ovarian cancer cells or MDA-MB-468 TNBC cells reduced the levels of p-TBK1, a marker for cGAS-STING activation (Fig. 10E). Together, these data support a role of mutp53 in the activation of cGAS-STING pathway in cancer cells.

Next, we queried cGAS-STING target gene expression among the patients who respond or do not respond to immune checkpoint inhibitors using the ROC Plotter server that includes all types of cancer. As shown in Fig. 10F and Fig. S11, the tumors that respond to immune checkpoint inhibitors express higher levels of cGAS-STING target genes. Using an unbiased pathway analysis in the ICBatlas server, we found that the top gene sets or pathways associated with a better response to immune checkpoint inhibitors are either cGAS-STING or its related pathways (Fig. 10G). GSEA analysis reveals a highly significant enrichment of the interferon-alpha response gene set in the responder group, irrespective of the cancer types or the immune checkpoint inhibitors used (Fig. 10H and Fig. S12). Lastly, we used Cancer-Immu server to analyze the response to immune checkpoint inhibitors in cancers harboring contact vs. conformational mutp53. Indeed, contact mutp53-harboring cancers have a better response to immune checkpoint inhibitors than conformational mutp53-harboring cancers (Fig. 10I, J). Taken together, these data show that contact mutp53 proteins enhance cGAS-STING activation in cancer and are linked to improved response to immune checkpoint inhibitors.

## Discussion
### Perturbation of DNA replication initiation by contact mutp53 activates the cGAS-STING pathway to enhance the response to immune checkpoint inhibition

Previously, we showed that some variants of contact mutp53 can override the requirement of Cdk2 to promote replication during G1/S transition by facilitating the TopBP1/Treslin interaction. Here we further demonstrate that contact mutp53 can perturb the replication initiation by interfering

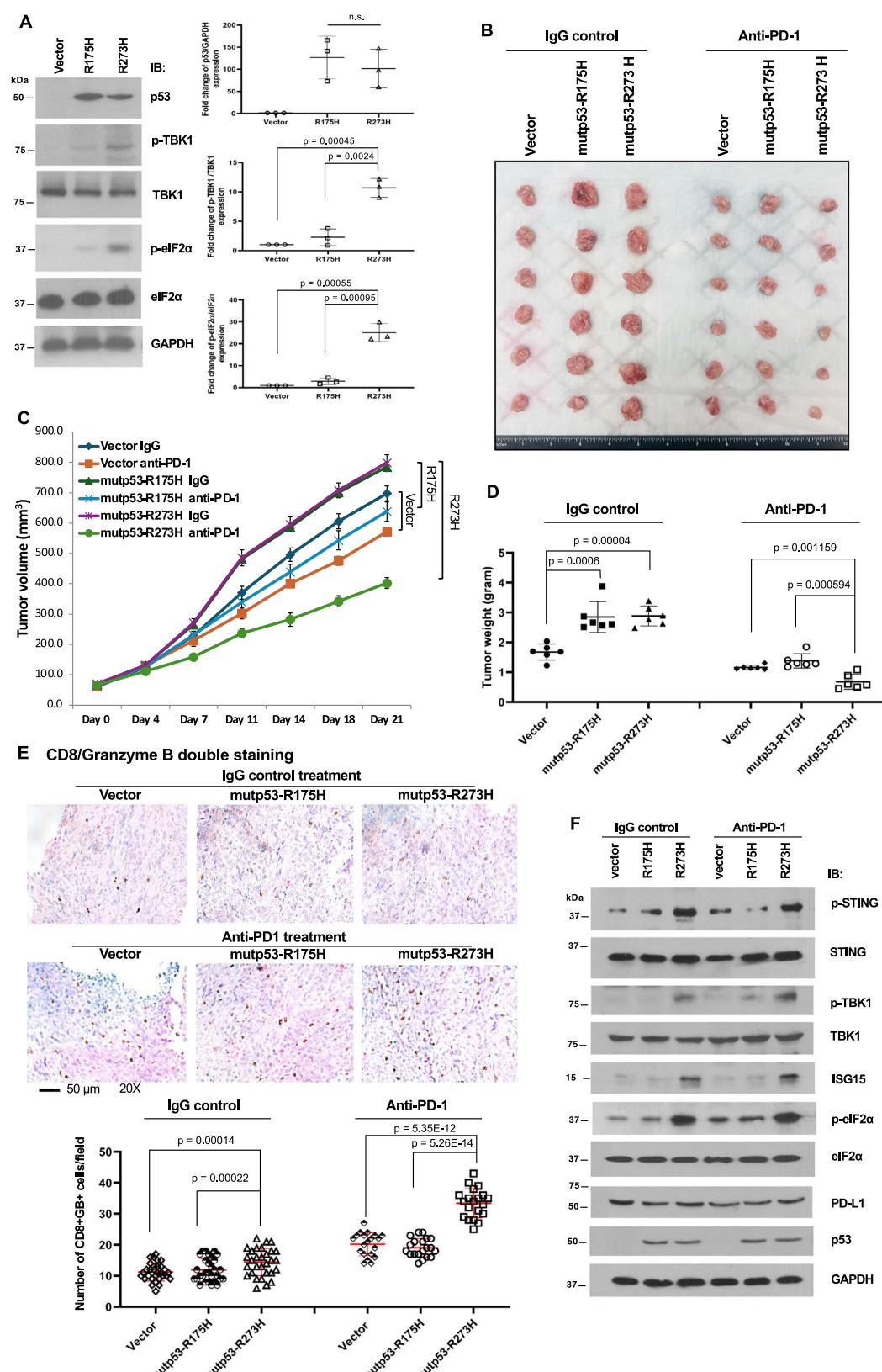

with the Akt-dependent switch of TopBP1 function in S and G2 phases. Normally, Akt is activated and phosphorylates TopBP1 during mid to late S phase, which results in the switch of TopBP1-interacting partner from Treslin to E2F1 (Fig. 10K). However, contact mutp53 can override this switch, allowing the persistent interaction of TopBP1 with Treslin, and therefore promotes replication initiation, even during the late S and G2 phases. Unregulated replication initiation can lead to re-replication, resulting in over-replicated regions, unequal distribution of DNA during cell division, and micronuclei formation. When DNA leaks from these aberrant regions or micronuclei to the cytoplasm, it is recognized as foreign

**Fig. 8 | Expression of mutp53-R273H, but not R175H, enhances the response to anti-PD-1 treatment in 4T1 mouse mammary allografts. A** The 4T1 mouse mammary tumor cells stably expressing a control vector, mutp53-R175H, or mutp53-R273H were harvested, and the whole cell lysates were subjected to immunoblotting with the indicated antibodies. The signal intensity of each protein was quantified using ImageJ software. The bar graph on the right represents the means and standard deviations obtained from three independent sets of stable cell lines (two-tailed $t$-test). **B–F** BALB/c mice bearing allografts of stable 4T1 cells described in A were treated with either control IgG isotype antibody or anti-PD-1 antibody (100 μg/mouse, twice weekly) via intraperitoneal injection for 3 weeks (n = 6 mice per group). **B** Representative photographs of the allograft tumors on the day of tumor harvesting. Tumor volume (**C**) and tumor weight (**D**) were measured and calculated. Data represent the means and standard deviations (two-tailed $t$-test). **E** Representative images of CD8 (brown)/Granzyme B (red) double staining at 20× magnification for each group. The number of positive double-stained cells per field was quantified from three allografts per group and shown in the graph below (mean ± SD, two-tailed $t$-test). **F** The whole cell lysates from a representative tumor in each allograft group were subjected to immunoblotting using the indicated antibodies.

or damaged DNA by the cGAS enzyme, leading to the activation of the cGAS-STING pathway. Our data provide evidence for the functional consequences of the perturbation of replication initiation by contact mutp53. Contact mutp53 promotes micronuclei formation, leading to the activation of the cGAS-STING pathway. Consequently, tumors harboring contact mutp53 show a greater response to immune checkpoint inhibition. Additionally, they respond more effectively to the combination of TopBP1-BRCT7/8 inhibitor CalAM with PARP inhibitors or ATR inhibitors.

### A previously unrecognized activity for contact mutp53 on DNA replication forks

During normal DNA replication, WT-p53 is localized to active replication forks and interacts with proteins involved in DNA replications[39]. p53 suppresses the activity of certain repair pathways that are prone to errors, such as those involving RAD52 and POLθ, and helps to maintain genomic integrity during replication[39]. Here, we describe a previously unrecognized activity for contact mutp53 on replication forks. The perturbation of DNA replication initiation is attributed to the ability of contact mutp53 to bind both Treslin and TopBP1 constantly during the S phase of the cell cycle. In contrast, conformational mutp53 can only bind TopBP1 but not Treslin. Although WT-p53 can also bind Treslin[6], its level is very low in S phase during normal proliferation, and does not perturb the regulation of TopBP1/Treslin interaction[3]. On the other hand, contact mutp53 frequently accumulates to very high levels due to the increased protein stability, and thus can perturb the Akt-dependent inhibition of DNA replication initiation.

### Differential activation of cGAS-STING by mutp53 variants influences the response to immune checkpoint inhibitors

WT-p53 can promote TRIM24-mediated degradation of TREX1, leading to the accumulation of cytosolic DNA, which in turn activates the cGAS-STING pathway to induce an antitumor immune response[27]. Although mutp53 loses the ability to regulate TRIM24-mediated degradation of TREX1, it can activate cGAS-STING through other mechanisms. Besides the perturbation of TopBP1/Treslin interaction as described in this study, some variants of mutp53 can bind certain MCM proteins and predispose cells to replication stress and chromosome instability[8]. Interestingly, unlike Treslin, MCM5 can bind to R175H, G245D, and R273H mutp53, but shows only weak interaction with WT-p53 or the R248Q mutp53 variant[8]. This suggests that mutp53 might influence DNA replication through multiple mechanisms. Further investigation is required to fully understand the impacts of various mutp53 variants on DNA replication. It is worth noting that in our experimental system using H1299 cells, we detected R273H mutp53 in nascent DNA, but not R175H (Fig. 3E). When compared side-by-side, mutp53-R273H exhibits a much stronger activity than mutp53-R175H (if at all) in cGAS-STING activation in both human H1299 and mouse 4T1 cell models (Figs. 7 and 8). The unique activation of cGAS-STING by mutp53-R273H, but not R175H or WT-p53, was further validated through a knockdown-reconstitution experiment in C33A cells (Fig. 7F). While both R175H and R273H mutp53 increased the growth of 4T1 allografts, only R273H enhanced the response to anti-PD-1 immunotherapy (Fig. 8).

Mutp53 has been reported to exert opposing effects on the cGAS-STING pathway[8,11]. On one hand, mutp53 interacts with TBK1 to suppress cGAS-STING-IRF3 signaling, thereby attenuating interferon responses and facilitating immune evasion[11]. On the other hand, in mutp53 cells, chromosomal instability activates cGAS-STING, leading to chronic NF-κB-driven inflammation that enhances tumor progression[8]. These findings highlight the context-dependent roles of mutp53 in modulating innate immune signaling. To ensure robustness, we examined the effects of mutp53 in multiple cell lines, including H1299, 4T1 (three independent sets of stable cell lines), C33A (knockdown-reconstitution), MDA-MB-468 (two independent sets of mutp53 knockdown cell lines), and MDA-MB-231, and in vivo 4T1 allograft models. We obtained consistent results throughout various cell lines and experimental settings. Our findings align with the study by Zhao et al.[8] and, importantly, reveal a differential impact of contact and conformational mutp53 on cGAS-STING activation and subsequent response to immune checkpoint inhibitors.

### Contact mutp53 activates cGAS-STING through synergistic MRE11 upregulation and TopBP1-driven micronuclei formation

While micronuclei formation alone is insufficient to activate cGAS-STING[28], MRE11 has been shown to release cGAS from nucleosome sequestration within micronuclei[15]. However, the pathological context in which both micronuclei formation and MRE11 activation co-occur remains unclear. Our study identifies contact mutp53 as a dual regulator that promotes both micronuclei formation and MRE11 expression, thereby enabling robust cGAS-STING activation (Fig. 10K). Previously, we reported that both contact and conformational mutp53 can promote the expression of NF-Y target genes[14]. Here, we further demonstrate that MRE11 is upregulated in mutp53-bearing cancers via the mutp53/TopBP1/NF-Y pathway. Importantly, depletion of TopBP1 or inhibition of DNA replication markedly reduces mutp53-R273H-induced cGAS-STING activation and p-eIF2α induction (Fig. 9G–J), strongly supporting the model in Fig. 10K.

### Limitations of the study

Different alleles of p53 mutants, even in the same class of contact or conformational mutants, can have distinct effects on transcriptional profiles and cellular behaviors, which may contribute to the clinical phenotypic heterogeneity at least in triple-negative breast tumors[40]. Our functional study focuses on two mutp53 variants, R175H and R273H. We also performed co-immunoprecipitation and confirmed that Treslin binds to two contact mutants R273H and R248W, but not to two conformational mutants R175H and R249S (Fig. 2F). This binding pattern correlates with their differential activities in activating the cGAS-STING pathway (Fig. 7E). Whether Treslin binds to every contact mutp53, but not conformational mutp53, remains to be determined in the future.

## Methods
### Cell culture
H1299, MDA-MB-468, MDA-MB-231, MDAH-2774, TOV-112D, and C33A cells were maintained in Dulbecco's modified Eagle's medium (DMEM) supplemented with 10% fetal bovine serum (FBS), SKBR3 cells were maintained in McCoy medium supplemented with 10% fetal bovine serum (FBS), and 4T1 cells were maintained in RPMI-1640 medium supplemented with 10% FBS. Penicillin (50 IU/ml) and streptomycin (50 μg/ml) were added to the culture medium. All cells were grown in a humidified

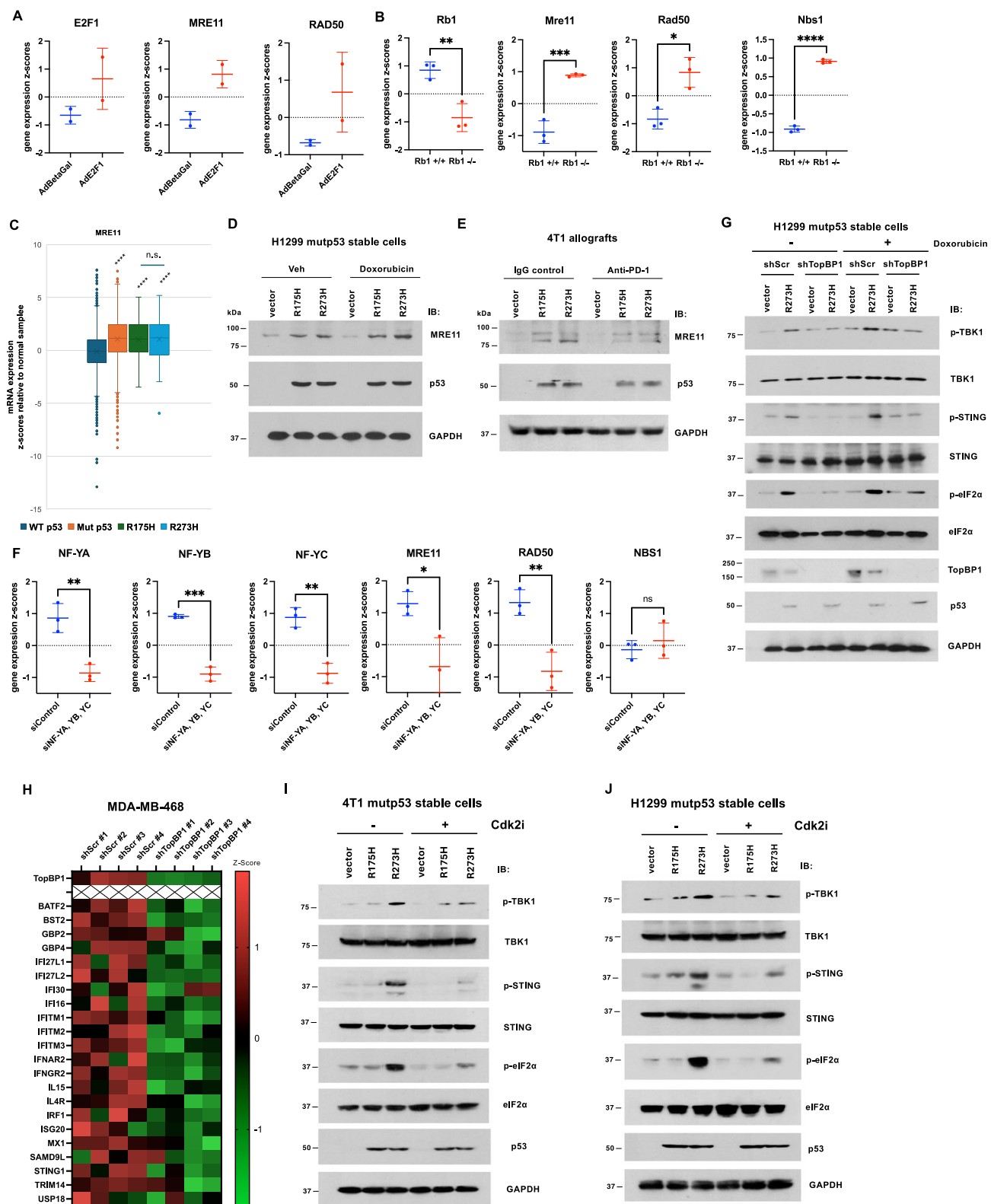

incubator at 37 °C with 5% $CO_2$ and 95% air. All cell lines were obtained from ATCC and tested negative for mycoplasma contamination.

## Establishment of stable cell lines

C33A, MDA-MB-468, or MDAH-2774 cell lines stably expressing mutp53 shRNA were established as described previously[14,23,41]. Briefly, the lentivirus carrying either pLKO.1-shScramble[42] or pLKO.1-shp53[43] was

produced in Lenti-X[TM] HEK293T (Takara), and was subsequently transduced into C33A, MDA-MB-468, or MDAH-2774 cells. After selection with puromycin (2 µg/ml for C33A; 3 µg/ml for MDA-MB-468 and MDAH-2774) for a week, stable cell lines were established for further experiments. To establish mutp53-harbring 4T1 cell lines, pCMV, pCMV-mutp53-R175H, or pCMV-mutp53-R273H was transfected into 4T1 cells, followed by selection with G418 (400 µg/ml). The effect of mutant p53 knockdown or

**Fig. 9 | Mutp53-R273H upregulates MRE11 and activates the cGAS-STING pathway in a TopBP1- and DNA replication-dependent manner.** MRN (MRE11-RAD50-NBS1) complex proteins are regulated by the Rb/E2F pathway. Gene expression data were extracted from GSE498[29] (**A**) and GSE54924[51] (**B**) through the NIH Gene Expression Omnibus (GEO) repository. In (**A**) an E2F1-expressing adenovirus was used to infect quiescent mouse fibroblasts, which barely express E2F1, and the gene expression profile was analyzed by Affymetrix microarray[29]. AdBetaGal (expressing β-galactosidase) serves as a control virus. In (**B**) total RNA in *Rb1*-knockout mouse embryonic fibroblasts was subjected to Affymetrix microarray analysis[51]. P-values were calculated using a two-tailed *t*-test. *P < 0.05, **P < 0.01, ***P < 0.001, ****P < 0.0001. **C** *MRE11* mRNA levels in a combined TCGA Pan-Cancer database, which contains 10967 samples from 32 studies compiled by cBioPortal. The data are stratified according to *TP53* status. N = 6281 (wild-type p53), 3520 (mutant p53), 121 (R175H), and 79 (R273H), respectively. ****P < 0.0001, compared to wild-type p53; n.s. not significant. **D, E** Expression of mutp53 enhances the expression of MRE11. **D** H1299 cells stably expressing an empty vector, mutp53-R175H or mutp53-R273H, were treated with vehicle control or Doxorubicin (800 nM) for 4 h. Cells were harvested, and lysates were subjected to immunoblotting. **E** Immunoblotting of the lysates from allografts described in Fig. 8 shows an induction of MRE11 by both mutp53-R175H and mutp53-R273H. **F** *MRE11* and *RAD50* mRNA levels are down-regulated upon NF-Y knockdown. Data shown are gene expression in mouse embryonic stem cells where NF-YA, YB, and YC were depleted. Data were obtained from GSE56840[52] via the NIH GEO repository. P-values were calculated using a two-tailed *t*-test. *P < 0.05, **P < 0.01, ***P < 0.001. **G** H1299 cells stably expressing either scrambled shRNA (shScr) or shTopBP1, along with an empty vector or mutp53-R273H[14], were treated with 800 nM Doxorubicin or not for 4 h. Cells were then harvested in SDS lysis buffer and subjected to SDS-PAGE, followed by immunoblotting with the indicated antibodies. **H** Heatmaps of cGAS-STING target gene expression in control versus TopBP1-depleted MDA-MB-468 (harboring mutp53-R273H). Gene expression data are deposited in GEO under accession number GSE227406[23]. 4T1 (**I**) and H1299 (**J**) stable cells harboring an empty vector, mutp53-R175H, or mutp53-R273H were treated with 2 µM CDK2 inhibitor II (Cdk2i) or not for 18 h. Cells were then harvested in SDS lysis buffer and subjected to SDS-PAGE, followed by immunoblotting with the indicated antibodies.

expression was confirmed by Western blotting using an antibody specific to p53.

## Immunoprecipitation and Western blot analysis

Cells were lysed in TNN buffer[21] (50 mM Tris, 0.25 M NaCl, 5 mM EDTA, 0.5% NP-40) supplemented with a protease inhibitor cocktail containing leupeptin (1 µg/ml), antipain (1 µg/ml), benzamidine (10 µg/ml), aprotinin (10 µg/ml), chymostatin (1 µg/ml), and pepstatin A (1 µg/ml). Immunoprecipitation was performed by incubating equal amounts of cell lysates with anti-TopBP1 mouse monoclonal antibody or control mouse IgG bound to protein A/G agarose beads (GenDepot). Myc-tagged fusion proteins in the whole cell lysates were immunoprecipitated with the Myc-Trap agarose beads. After incubation at 4 °C for 16 h, immunoprecipitates were washed three times, followed by fractionation with SDS-PAGE and electrotransferred to Imobilon-P membrane (Millipore). Immunoblotting was performed with appropriate antibodies. For chromatin immunoprecipitation of BrdU-labeled DNA and its associated complex, chromatin was sonicated as previously described[44] and then immunoprecipitation was performed with anti-BrdU antibody (Clone B44, BD Biosciences), followed by immunoblotting.

## Double thymidine block

TOV-112D, MDAH-2774, SKBR3, MDA-MB-468, H1299, and C33A cells were treated with 2 mM thymidine for 18 h. After washing twice in phosphate-buffered saline (PBS), cells were cultured in fresh medium with 10% FBS for 9 h and then treated with 2 mM thymidine for another 18 h. After washing twice with PBS, cells were incubated in medium containing 10% FBS and then collected at the designated time points for assays.

## Flow cytometry

To determine the DNA content profiles of synchronized cells, a portion of the cells was fixed with 70% ethanol and then stained with propidium iodide. Flow cytometry was performed in the BCM cytometry and cell sorting facility on a BD FACSCanto II. Equivalent numbers (at least 10,000) of cells per sample were assayed. Data analysis, including cell cycle modeling, was performed using FlowJo software.

## Replication labeling and DNA fiber spread assay

H1299 cells were transfected with an empty vector (pCMV), pCMV-mutp53-R175H, or -R273H. After 24 h, cells were subjected to the double thymidine block procedure as described above. After release, cells were labeled with 50 µM IdU at the designated time points for 30 min, and then washed with PBS, followed by incubation with 250 µM CIdU for another 30 min. DNA spread assay was performed as previously described[16] with some modifications. Briefly, cells were harvested and resuspended in

ice-cold PBS at ~5 × 10⁶ cells/ml. Two µl of the cell suspension were spotted at the upper end of the microscope slide (Silane-Prep slides, S4651-72EA from Sigma) and air-dried for ~5 min. Subsequently, 7 µl of lysis buffer (200 mM Tris-HCl, pH 7.5, 50 mM EDTA, and 0.5% SDS) was applied on top of the cell suspension. Two minutes later, the slides were tilted to 15° to allow the spread of DNA fibers. Slides were air-dried, fixed with methanol/acetic acid (3:1) for 10 min, and refrigerated overnight. DNA was denatured with 2.5 M HCl for 80 min at room temperature. The slides were rinsed three times in PBS and incubated in blocking buffer (5% BSA in PBS) for 30 min. 150 µl of the primary antibodies in blocking solution (1:200 anti-BrdU antibody [mouse, from BD Biosciences], 1:400 anti-BrdU antibody [rat, from Abcam]) were pipetted on each slide, the slides were gently covered with a coverslip and incubated for 2 h at room temperature. Subsequently, coverslips were removed, and slides were washed three times with PBS. 150 µl of the secondary antibodies in blocking solution (1:400 goat anti-mouse Texas-Red-X), 1:400 chicken anti-rat Alexa Fluor 488 were applied on each slide, and the slides were covered again with a coverslip for 1 h. Following the removal of the coverslip and three times PBS washing, the slides were applied with coverslips, mounted, and stored at −20 °C before analysis. Images were captured with a Zeiss fluorescence microscope (Axio Observer Inverted Microscope).

## Cytokinesis-block micronucleus (CBMN) assay

H1299 cells transfected with pCMV, pCMV-mutp53-R175H, or pCMV-mutp53-R273H were seeded on coverslips in 6-well plates. 24 h later, cells were treated or not with different doses of drugs for 4 h. After PBS washing three times, Cytochalasin B (3 µg/ml; Sigma-Aldrich) was added to the medium for 24 h to block cytokinesis. Cells were then fixed with 4% Formaldehyde for 12 h and stained with bisBenzimide (Sigma). The number of micronuclei (MN) in each binucleated (BN) cell was counted under a microscope. Approximately 400 BN cells per slide were scored. The experiment was done in triplicate. Images were captured with a Zeiss fluorescence microscope (Axio Observer Inverted Microscope).

## Cell counting Kit-8 (CCK-8) assay

Cell viability was determined by cell counting kit-8 (CCK-8) assay according to the manufacturer's instructions (APExBIO). Briefly, H1299 cells were transfected with an empty vector (pCMV), pCMV-mutp53-R175H, or -R273H. After 24 h, equal amounts of cells were seeded on 48-well plates for 24 h. Cells were treated or not with various concentrations of drugs for 48 h. Ten µl of WST-8 solution was added to the cells for one hour at 37 °C, and the amount of WST-8 formazan generated by the activity of dehydrogenases in the living cells was measured by reading the absorbance at OD 450 nm using a microplate reader (BioTek Synergy HT). Each experiment was performed at least in triplicate.

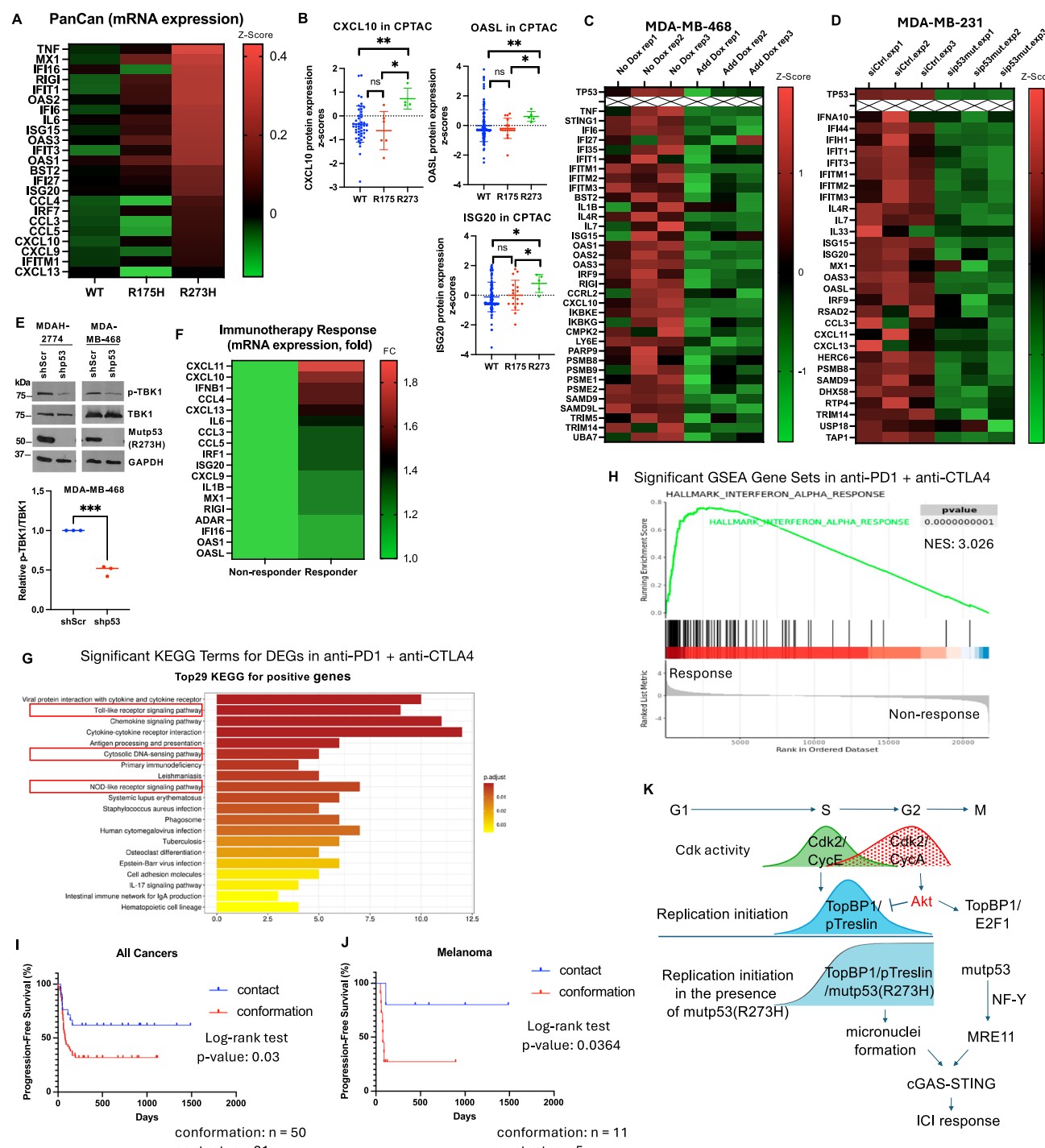

## Real-time RT-PCR

H1299 cells were transfected with pCMV, pCMV-mutp53-R175H, or pCMV-mutp53-R273H. After 48 h, cells were treated or not with 800 nM Doxorubicin for 4 h. RNA was extracted using TRIzol reagent (Invitrogen). Reverse transcription-quantitative PCR (RT-qPCR) was performed using the appropriate primer pairs.

## In vivo allograft experiments

Stable 4T1 cell lines harboring an empty vector, mutp53-R175H, or mutp53-R273H (0.5 million cells per site in 100 μl PBS) were injected subcutaneously into the right side of the flank of the 6-week-old female BALB/c mice. Since 4T1 cells were derived from breast cancer, which

predominantly affects females, only female mice were used to ensure the relevance and applicability of the findings. When tumors were measurable, mice were randomized into two groups and injected intraperitoneally with anti-mouse CD279 (PD-1) antibody (clone 29F.1A12) or Rat IgG$_{2a}$ isotype control at 100 μg/mouse, twice per week. The mouse weight and tumor size were measured twice per week with a caliper and calculated based on the formula: Tumor volume = (π/6) × (length × depth × width). The evaluator was blinded to the group allocation during monitoring. Animals were euthanized on the indicated dates, and tumors were harvested, weighed, and further processed for histopathological analysis. All experiments were performed under a Baylor College of Medicine Institutional Animal Care and Use Committee (IACUC)-approved protocol, and all experiments conform

**Fig. 10 | Contact mutations of *TP53* are associated with cGAS-STING activation and better response to immune checkpoint inhibitor therapy in multiple types of cancer. A** Heatmap of mRNA expression of cGAS-STING target genes in tumors that harbor either wild-type (WT) *TP53* or R175H or R273H mutations in the TCGA Pan-Cancer database. Shown are mean z-scores relative to diploid samples. The statistical analysis of each individual gene is shown in Fig. S7. **B** The protein levels of CXCL10, OASL, and ISG20 in tumors that harbor either wild-type (WT) *TP53* or mutations in R175 or R273 residue in the TCGA CPTAC database. *$P < 0.05$, **$P < 0.01$, ns not significant (two-tailed *t*-test). **C, D** Heatmaps of cGAS-STING target gene expression in MDA-MB-468 or MDA-MB-231 in which mutp53 was depleted. Gene expression data were extracted from GSE31812[37] and GSE26262[38]. In MDA-MB-468 cells, the mutp53 shRNA was induced by doxycycline (Dox). **E** Depletion of mutp53 significantly decreases TBK1 phosphorylation. MDAH-2774 or MDA-MB-468 cells stably expressing either a scrambled shRNA or a mutp53 shRNA were harvested, and the total cell lysates were subjected to immunoblotting using antibodies specific to p-TBK1, TBK1, p53, or GAPDH. Lower panel: Relative p-TBK1/TBK1 levels were quantified from three independent experiments using two distinct sets of MDA-MB-468 cells stably expressing a scrambled shRNA or a mutp53 shRNA. **F** Heatmap of cGAS-STING target gene expression in pretreatment tumors stratified according to their response to immune checkpoint inhibitors. Shown are mean fold changes (Responders/Non-responders) for each gene. The boxplots of each gene are shown in Fig. S11. Analysis was performed in ROC Plotter, including all types of cancer and any immune checkpoint inhibitors used with n = 1267 (809 non-responders and 458 responders). **G, H** Pathway and GSEA analyses of DEGs (differentially expressed genes) in the responders vs. non-responders to anti-PD-1 + anti-CTLA4. Datasets from melanoma patients (Accession number: ERP105482) and renal cell carcinoma patients (Accession number: SRP128156) were analyzed in the ICBatlas server. N = 13 (non-responders) and 25 (responders). H: GSEA of Hallmark_Interferon_alpha_response; NES: normalized enrichment score. **I, J** Kaplan–Meier curves of progression-free survival (PFS) after immune checkpoint inhibitor therapy in patients with cancers harboring either contact or conformational mutp53. PFS and response to immune checkpoint inhibitors were accessed from Pan-cancer analysis in Cancer_Immu server. Mutations of *TP53* were retrieved from the cBioPortal server. **K** Schematic shows how mutp53-R273H perturbs the tight control of DNA replication initiation at late S-G2 phases and leads to persistent replication initiation that causes accumulation of micronuclei. Mutp53 can also induce MRE11 through NF-Y. The combination of micronuclei formation and elevated MRE11 expression activates the cGAS-STING pathway and renders cancers more responsive to immune checkpoint inhibitors.

to IACUC standards and ethical regulations. We have complied with all relevant ethical regulations for animal use. Sample size was based on estimations by power analysis with a level of significance of 0.05 and a power of 0.9.

### Histology and immunohistochemistry
Tumor samples were fixed in 10% neutral buffered formalin and sent to the Pathology and Histology Core of Baylor College of Medicine for further processing. The paraffin-embedded sections were stained with anti-CD8 and anti-Granzyme B antibodies using standard operating protocols[45]. CD8 was stained with 3,3′-Diaminobenzidine (DAB) (brown) and granzyme B with alkaline phosphatase-fast red (AP/Red) (red) to produce a red-brown color combination.

### Statistics and reproducibility
Two-tailed *t*-test was performed to compare two experimental groups. One-way ANOVA was used to determine whether three or four groups are statistically different from each other. Data were presented as means ± S.D. from at least three biological replicates. *P*-values less than 0.05 were considered statistically significant. RNA-Seq, protein expressions, and *TP53* mutations in the TCGA Pan-Cancer database were extracted from the cBioPortal server. Gene expression in GSE31812 and GSE26262 was retrieved in the NCBI Gene Expression Omnibus (GEO). The gene expression values were transformed to z-scores for heatmap display using the formula: $Z = (X - \mu)/\sigma$. Where: Z is the z-score; X is the gene expression value; μ is the mean of the dataset; and σ is the standard deviation of the dataset. Clinical data of the response to immune checkpoint inhibitors were retrieved from Pan-cancer analysis in Cancer_Immu server, and mutations of *TP53* in these tumors were retrieved through cBioPortal. Based on the literature[13,46-50], contact mutp53 variants such as R248W/Q, R273H/C/L, and R280K, and conformational mutp53 variants including V143A/M, V157F, R175H, C176F/Y, H179R/Y, M237I, G245S/D, R249S, Y205C, Y220C, Y163C, Y234C and Y236C were included in this study.

All key resources (including antibodies, cell lines, mouse strain, oligonucleotides, and plasmids) are listed in Supplementary Table S1.

### Reporting summary
Further information on research design is available in the Nature Portfolio Reporting Summary linked to this article.

### Data availability
All study data, uncropped blots, and numerical source data are included in the article, Supporting Information, and Supplementary Data 1 and 2. This paper does not report original code.

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

## Acknowledgements

This work was supported by funding from NIH R01CA203824, R01CA269971, Department of Defense Grants W81XWH-18-1-0329, W81XWH-19-1-0369, W81XWH-22-1-0226, W81XWH-22-1-0534, HT9425-24-1-0045 (to W.-C.L. and F.-T.L.), and Rivkin Center for Ovarian Cancer Pilot Award (to W.-C.L.). L.A.W.G. was supported by T32GM136560. We also acknowledge the support from the Cytometry and Cell Sorting Core and Pathology & Histology Core at BCM.

## Author contributions

W.-C.L. conceived the study; K.L. and W.-C.L. designed the research; K.L. performed most experiments; L.A.W.G. performed BrdU/PI flow cytometry analysis; F.-T.L. performed some experiments; W.-C.L. performed bioinformatics analyses; K.L., L.A.W.G., F.-T.L., and W.-C.L. analyzed data; and K.L., F.-T.L., and W.-C.L. wrote the paper. W.-C.L. and F.-T.L. acquired funding. All co-authors contributed to the final version of the manuscript.

## Competing interests

W.-C.L. and F.-T.L. are co-founders of ETIEN Bio, Inc. All other authors declare no competing interests.
