## [Transparent Peer Review file · Communications Biology]

Mutant p53 variants differentially impact replication initiation and activate cGAS-STING to affect immune checkpoint inhibition

Corresponding Author: Professor Weei-Chin Lin

Version 0:

Reviewer comments:

Reviewer #1

(Remarks to the Author)

This manuscript submission entitled “Mutant p53 variants differentially impact replication and activate cGAS-STING to affect immune checkpoint inhibition” presents a number of findings that will be of interest to the mutant p53 biology field but may not be of strong general interest. The current title is written in vague and passive format. As such the title of this manuscript needs correction to articulate the findings of the research. The authors demonstrate the interesting finding that contact, but not conformational, missense mutant p53 proteins activate replication initiation and the cGAS-STING pathway. The authors find that when contact mutant p53 R273H is expressed in 4T1 cells it increases tumorigenesis that can be targeted by immune checkpoint inhibition. The data presented are clear and will be of interest to others in the community. However, there is a conceptual flaw as the authors fail to acknowledge the previous work in the mutant p53 field with respect to findings already shown for the impact of mutant p53 on the cGAS-STING pathway in different breast cancer cells including 4T1 cells. As such, some of the findings are original while others are not. Moreover, the introduction fails to clearly articulate the gap in knowledge in the context for what has already been published for mutant p53 expression in 4T1 cells in xenograft experiments, as well as how mutant p53 drives tumorigenesis, and that combination treatments with PARP inhibitors targeted at different breast cancers expressing mutant p53 R273H has been previously demonstrated to increase sensitivity for cells with mutant p53 associated with replicating DNA. The conclusion that is the most original is not that mutant p53 can activate the cGAS-STING pathway, but rather that the conformation missense mutant p53 proteins do not do so and do not associate with increased initiation of DNA replication and micronuclei formation. The original conclusions must be placed in better context. Some relevant references that should be credited for their findings in context to mutant p53 are included in the context of their findings below. While some of the references are included they are not described in the context of what they contributed and how the findings in this submission expands upon them.

The contact mutant p53 interaction with the MCM helicase DNA replication pathway allows for a chronic chromosomal-instability (CIN) state that leads to a low-level cytosolic DNA response that promotes pro-tumorigenic inflammation(1). Moreover contact mtp53 hinders TNBC ferroptotic cell death to promote cell survival (2), and contact mtp53 expression inhibits the cGAS/STING-induced interferon type I (IFN-I) pathway activation reducing anti-tumor immunity (3). PARP and contact mutant p53 interact together at replication forks, and PARP inhibitor talazoparib treatment in combination with temozolomide can kill missense mutant p53 expressing breast cancer cells in culture (4) and in patient derived tumor organoids (5).

The work presents convincing evidence for the difference in cGAS-STING activation between contact and conformational missense mutant p53. Therefore this should be the focus of the manuscript title, introduction, and discussion. A great deal of previous work established many of the findings shown for the contact mutants but the gap in knowledge addressed in the submitted manuscript was not well articulated to describe that what was new was the comparison between contact and conformational mutants. As such the introduction and discussion need to be revised to better introduce what is new in this paper and what is not. No further experimental evidence is required, however to strengthen the conclusions the researchers must frame the gap in knowledge as the comparison of different forms of missense mutant p53 expression on the outcomes to potential therapies that have only been tested in the p53 contact mutant context previously. Such revised framing of the

question, along with a more pointed introduction about the questions being addressed, will assist the paper in influence thinking in the field to carefully consider how different mutant p53 structures influence genomic instability and potentially treatment paradigms.

1. Zhao, M., Wang, T., Gleber-Netto, F.O., Chen, Z., McGrail, D.J., Gomez, J.A., Ju, W., Gadhikar, M.A., Ma, W., Shen, L. et al. (2024) Mutant p53 gains oncogenic functions through a chromosomal instability-induced cytosolic DNA response. *Nat Commun*, 15, 180.
2. Dibra, D., Xiong, S., Moyer, S.M., El-Naggar, A.K., Qi, Y., Su, X., Kong, E.K., Korkut, A. and Lozano, G. (2024) Mutant p53 protects triple-negative breast adenocarcinomas from ferroptosis in vivo. *Sci Adv*, 10, eadk1835.
3. Ghosh, M., Saha, S., Bettke, J., Nagar, R., Parrales, A., Iwakuma, T., van der Velden, A.W.M. and Martinez, L.A. (2021) Mutant p53 suppresses innate immune signaling to promote tumorigenesis. *Cancer Cell*.
4. Xiao, G., Lundine, D., Annor, G.K., Canar, J., Ellison, V., Polotskaia, A., Donabedian, P.L., Reiner, T., Khramtsova, G.F., Olopade, O.I. et al. (2020) Gain-of-Function Mutant p53 R273H Interacts with Replicating DNA and PARP1 in Breast Cancer. *Cancer Res*, 80, 394-405.
5. Madorsky Rowdo, F.P., Xiao, G., Khramtsova, G.F., Nguyen, J., Martini, R., Stonaker, B., Boateng, R., Oppong, J.K., Adjei, E.K., Awuah, B. et al. (2024) Patient-derived tumor organoids with p53 mutations, and not wild-type p53, are sensitive to synergistic combination PARP inhibitor treatment. *Cancer Lett*, 584, 216608.

Reviewer #2

(Remarks to the Author)

Review of Liu et. al.

The manuscript by Liu et al focuses on the role of the TopBP1/Treslin complex in regulating the initiation of DNA replication. The manuscript provides some evidence suggesting that contact mutants of p53 that cannot bind DNA preserve the TopBP1/Treslin complex, leading to the initiation of DNA replication in late S and G2 phases. This results in micronuclei formation and presumably genomic instability. They go on to demonstrate that inhibition of the TopBP1/Treslin interaction synergises with inhibition of either ATR or PARP leading to decreased cell proliferation. Finally, they demonstrate that the contact mutants induce cGAS/STING, leading to sensitisation of tumours to immune checkpoint blockage. While the data in the paper are interesting, there are some concerns that need to be addressed before the manuscript can be accepted for publication.

Major comments

1. The authors claim that the presence of the contact mutants of p53 prevent the switch from TopBP1/Treslin to TopBP1/E2F1. However, what is the evidence that all of TopBP1 is phosphorylated by Akt in this assay? Might there be two pools, a phosphorylated and an unphosphorylated pool? I would have liked to see a similar result in cells that had the mutant p53 KD and WT p53 re-expressed to see the difference, because it looks like TopBP1 still retains binding to E2F1 in their assays once Akt is activated. Therefore, is it really a switch? Is TopBP1 or Treslin stabilised in the presence of R273H? This might provide an alternative explanation for their results. These concerns need to be addressed.
2. While I am reasonably convinced that R273H leads to reduplication and micronuclei formation and to activation of c-GAS/STING, I do not see any correlation between these two phenotypes. I would have loved to see an experiment where TopBP1 is knocked down in R273H cells, and see if this leads to a decrease in c-GAS/STING. This would be really interesting and add significantly to the paper.

Minor comments.

1. I did not see any experiments where the mutants were compared to WT p53. They would have added significantly to the manuscript. Also, I would have liked to see experiments where the endogenous mutant p53 was knocked down in breast cancer lines and replaced with the WT or mutant p53 proteins thus eliminating the contribution of other changes to the phenotypes observed.

Version 1:

Reviewer comments:

Reviewer #2

(Remarks to the Author)

The authors have addressed some of my criticisms, which improve the manuscript. However, I cannot find the experiments that they have listed in their response to the review. I quote "Since re-expression of WT p53 would arrest the cells and make cell cycle synchronization impossible, we performed mutp53 knockdown in C33A (Fig. 3D-E). Indeed, depletion of mutp53 rescued the inhibitory regulation of TopBP1/Treslin interaction in mid-late S phase (Fig. 3D-E), supporting a role for mutp53

in bridging

TopBP1/Treslin interaction in mid-late S phase."

The C33A knockdown experiments are in figure 7. I see no experiments where the mutant p53 is knocked down followed by a demonstration of the interaction between TopBP1/Treslin. Figure 3D-E are DNA fiber length measurements.

This needs to be corrected by the authors before the manuscript can be accepted.

We greatly appreciate the reviewers' comments. We have extensively revised the manuscript and performed additional experiments (**Fig. 7F, 9G, 9H, 9I, 9J, and 10E**) to address all of the reviewers' concerns.

Point-by-point response:

Reviewer #1:

This manuscript submission entitled “Mutant p53 variants differentially impact replication and activate cGAS-STING to affect immune checkpoint inhibition” presents a number of findings that will be of interest to the mutant p53 biology field but may not be of strong general interest. The current title is written in vague and passive format. As such the title of this manuscript needs correction to articulate the findings of the research. The authors demonstrate the interesting finding that contact, but not conformational, missense mutant p53 proteins activate replication initiation and the cGAS-STING pathway. The authors find that when contact mutant p53 R273H is expressed in 4T1 cells it increases tumorigenesis that can be targeted by immune checkpoint inhibition. The data presented are clear and will be of interest to others in the community. However, there is a conceptual flaw as the authors fail to acknowledge the previous work in the mutant p53 field with respect to findings already shown for the impact of mutant p53 on the cGAS-STING pathway in different breast cancer cells including 4T1 cells. As such, some of the findings are original while others are not. Moreover, the introduction fails to clearly articulate the gap in knowledge in the context for what has already been published for mutant p53 expression in 4T1 cells in xenograft experiments, as well as how mutant p53 drives tumorigenesis, and that combination treatments with PARP inhibitors targeted at different breast cancers expressing mutant p53 R273H has been previously demonstrated to increase sensitivity for cells with mutant p53 associated with replicating DNA. The conclusion that is the most original is not that mutant p53 can activate the cGAS-STING pathway, but rather that the conformation missense mutant p53 proteins do not do so and do not associate with increased initiation of DNA replication and micronuclei formation. The original conclusions must be placed in better context. Some relevant references that should be credited for their findings in context to mutant p53 are included in the context of their findings below. While some of the references are included they are not described in the context of what they contributed and how the findings in this submission expands upon them.

The contact mutant p53 interaction with the MCM helicase DNA replication pathway allows for a chronic chromosomal-instability (CIN) state that leads to a low-level cytosolic DNA response that promotes pro-tumorigenic inflammation(1). Moreover contact mtp53 hinders TNBC ferroptotic cell death to promote cell survival (2), and contact mtp53 expression inhibits the cGAS/STING-induced interferon type I (IFN-I) pathway activation reducing anti-tumor immunity (3). PARP and contact mutant p53 interact together at replication forks, and PARP inhibitor talazoparib treatment in combination with temozolomide can kill missense mutant p53 expressing breast cancer cells in culture (4) and in patient derived tumor organoids (5).

The work presents convincing evidence for the difference in cGAS-STING activation between contact and conformational missense mutant p53. Therefore this should be the focus of the manuscript title, introduction, and discussion. A great deal of previous work established many of the findings shown for the contact mutants but the gap in knowledge addressed in the submitted manuscript was not well articulated to describe that what was new was the comparison between

contact and conformational mutants. As such the introduction and discussion need to be revised to better introduce what is new in this paper and what is not. No further experimental evidence is required, however to strengthen the conclusions the researchers must frame the gap in knowledge as the comparison of different forms of missense mutant p53 expression on the outcomes to potential therapies that have only been tested in the p53 contact mutant context previously. Such revised framing of the question, along with a more pointed introduction about the questions being addressed, will assist the paper in influence thinking in the field to carefully consider how different mutant p53 structures influence genomic instability and potentially treatment paradigms.

Response: We appreciate the reviewer's thoughtful comments. In response, we have substantially revised both the Introduction and Discussion to better contextualize prior work in the mutant p53 field, particularly regarding its impact on the cGAS-STING pathway. We now emphasize the knowledge gap between contact and conformational p53 mutants. Specifically, we highlight the differential impact of different mutant p53 forms on therapeutic outcomes, noting that prior studies have largely focused on contact mutants and have not examined the therapeutic outcomes. Our revised manuscript includes a direct comparison between contact and conformational mutants, and we have incorporated the additional references suggested by the reviewer.

We respectfully disagree, at least in part, with the comment that “some of the findings are original while others are not.” We would like to clarify that all data presented in this manuscript are original and distinct from those in our previous publication (including Liu et al., PNAS, 2017). Conceptually, the prior study focused on **Cdk2**-regulated **G1/S** transition, whereas the current manuscript investigates the impact of mutp53 on **Akt**-mediated inhibition of the TopBP1/Treslin interaction during **late S and G2 phases**. While some methodologies are shared, the scientific questions and conclusions are distinct.

We acknowledge that activation of cGAS-STING by mutp53 has been reported and cited in our original manuscript. However, our study delineates a distinct mechanistic pathway—specifically involving Treslin, in contrast to the MCM proteins described in prior work. Furthermore, our investigation compares contact and conformational mutp53 variants side-by-side, evaluating their differential impact on the response to immune checkpoint inhibitor therapy, an aspect not addressed in earlier studies. In this revision, we have discussed the published studies and positioned our new findings in relation to previously published studies, as this reviewer kindly suggested.

Reviewer #2:

The manuscript by Liu et al focuses on the role of the TopBP1/Treslin complex in regulating the initiation of DNA replication. The manuscript provides some evidence suggesting that contact mutants of p53 that cannot bind DNA preserve the TopBP1/Treslin complex, leading to the initiation of DNA replication in late S and G2 phases. This results in micronuclei formation and presumably genomic instability. They go on to demonstrate that inhibition of the TopBP1/Treslin interaction synergises with inhibition of either ATR or PARP leading to decreased cell

proliferation. Finally, they demonstrate that the contact mutants induce cGAS/STING, leading to sensitisation of tumours to immune checkpoint blockage. While the data in the paper are interesting, there are some concerns that need to be addressed before the manuscript can be accepted for publication.

Major comments

1. The authors claim that the presence of the contact mutants of p53 prevent the switch from TopBP1/Treslin to TopBP1/E2F1. However, what is the evidence that all of TopBP1 is phosphorylated by Akt in this assay? Might there be two pools, a phosphorylated and an unphosphorylated pool? I would have liked to see a similar result in cells that had the mutant p53 KD and WT p53 re-expressed to see the difference, because it looks like TopBP1 still retains binding to E2F1 in their assays once Akt is activated. Therefore, is it really a switch? Is TopBP1 or Treslin stabilised in the presence of R273H? This might provide an alternative explanation for their results. These concerns need to be addressed.

Response: We appreciate the reviewer's suggestion. We have performed the mutant p53 KD and WT p53 re-expression experiment to look at the effect on cGAS-STING and present the result in **Fig. 7F**, which is discussed in the last response. Since re-expression of WT p53 would arrest the cells and make cell cycle synchronization impossible, we performed mutp53 knockdown in C33A (**Fig. 3D-E**). Indeed, depletion of mutp53 rescued the inhibitory regulation of TopBP1/Treslin interaction in mid-late S phase (Fig. 3D-E), supporting a role for mutp53 in bridging TopBP1/Treslin interaction in mid-late S phase.

The reviewer is correct that there are two pools of TopBP1 inside the cells, an Akt-phosphorylated and an Akt-unphosphorylated pool. Upon Akt phosphorylation, **all** TopBP1 that has been phosphorylated by Akt at S1159 (detected by anti-pS1159-TopBP1 antibody) forms oligomers, and binds E2F1. The "switch" from monomers to oligomers by Akt phosphorylation is supported by a size exclusion chromatography experiment using purified TopBP1-BRCT6/7/8 protein (a selected data from PMID: 24081328 is shown below).

In our subsequent paper (PMID: 31964753), we show that Treslin can only bind Akt-unphosphorylated, but not Akt-phosphorylated, TopBP1. The levels of Akt-phosphorylated TopBP1 is very low level in G1 phase, but rises sharply at mid S phase to G2 phase, at which time all TopBP1 is associated with E2F1, but not Treslin, supporting that most of TopBP1 is in the Akt-phosphorylated form during mid-S and G2 phases (PMID: 31964753). Therefore, there are ample evidence to support the switch from TopBP1/Treslin to TopBP1/E2F1 during late G1 transition to S/G2 phases (PMID: 24081328, 31964753).

We do agree with the reviewer that while mutp53-R273H makes TopBP1/Treslin interaction appear throughout S/G2 phases, it does not affect TopBP1/E2F1 interaction. So, in the presence of mutp53-R273H, during mid-late S phase, we detected both TopBP1/Treslin and TopBP1/E2F1 complexes. To more precisely describe the effect of mutp53, we revised “the differential effect of mutp53 variants on the switch of TopBP1 binding from Treslin to E2F1” to “the differential effect of mutp53 variants on the control of TopBP1/Treslin binding”, and revised “depletion of mutp53 restored the switch of TopBP1 binding from Treslin to E2F1 during S phase progression” to “depletion of mutp53 restored the regulation of TopBP1/Treslin binding during S phase progression”.

The expression of Treslin was not changed upon coexpression with various forms of mutp53, regardless whether binding or not (see Fig. 2F, input western blots). Coexpression of mutp53-R273H with Treslin or TopBP1 also did not affect the expression of Treslin or TopBP1 (Fig. 5C in PMID: 28439015; Fig. 1A in PMID: 21930790). Therefore, we do not think TopBP1 or Treslin is stabilized in the presence of R273H.

2. While I am reasonably convinced that R273H leads to reduplication and micronuclei formation and to activation of c-GAS/STING, I do not see any correlation between these two phenotypes. I would have loved to see an experiment where TopBP1 is knocked down in R273H cells, and see if this leads to a decrease in c-GAS/STING. This would be really interesting and add significantly to the paper.

Response: We appreciate the constructive suggestion from this reviewer. We have performed TopBP1-knockdown experiments and presented the new data in **Fig. 9G and H**. The new data show that the effect of R273H on cGAS-STING activation is abrogated upon TopBP1 depletion in both H1299 and MDA-MB-468 cells. We also used Cdk2 inhibitor to inhibit DNA replication and show that the activation of cGAS-STING by R273H is dependent on active DNA replication (**Fig. 9I-J**). Thus, these new data provide a strong link between reduplication and micronuclei formation and activation of c-GAS/STING.

Minor comments.

1. I did not see any experiments where the mutants were compared to WT p53. They would have added significantly to the manuscript. Also, I would have liked to see experiments where the endogenous mutant p53 was knocked down in breast cancer lines and replaced with the WT or mutant p53 proteins thus eliminating the contribution of other changes to the phenotypes observed.

Response: We include a comparison with WT-p53 in **Fig. 7E**, which demonstrates that only contact mutp53, but not WT or conformation mutp53, can activate cGAS-STING. We also have performed a new knockdown/reconstitution experiment as suggested by this reviewer and presented the data in **Fig. 7F**. The new data further confirm the distinct ability of R273H, in contrast to WT-p53 and R175H, to activate the cGAS-STING pathway.

We compared mutp53 with WT-p53 in their binding to Treslin, as shown in **Fig. 2F**. Similarly to WT-p53, contact mutp53 retains the ability to bind Treslin; however, conformational mutp53 loses

this binding capacity. As we discuss in the Discussion section (Line 403-407): “*although WT-p53 can also bind Treslin⁶, its level is very low in S phase during normal proliferation, and does not perturb the regulation of TopBP1/Treslin interaction³. On the other hand, mutant mutp53 frequently accumulates to very high levels due to the increased protein stability, and thus can perturb the Akt-dependent inhibition of DNA replication initiation.*” In H1299 cells overexpressing WT-p53 (**Fig. 7E**), the induction of cell cycle arrest or apoptosis is expected to inhibit DNA replication, thereby preventing activation of the cGAS-STING pathway through the replication-mediated mechanism. This is consistent with the effect of a Cdk2 inhibitor shown in **Fig. 9I-J**. We did observe a modest effect of WT-p53 on cGAS-STING activation especially in shScr control C33A (harboring mutp53-R273C) (**Fig. 7F**). This observation is consistent with a prior report about WT-p53 and we have incorporated the following statement to the revised manuscript (Line 263-267): “*WT-p53 can facilitate the degradation of cytosolic DNA exonuclease TREX1, leading to the accumulation of cytosolic DNA and subsequent activation of the cGAS-STING pathway²⁷. It is plausible that due to mutp53-R273C expression, shScr control C33A cells harbor elevated levels of cytosolic DNA, thereby amplifying the effect of WT-p53 on cGAS-STING activation in this context.*”

Point-by-point response:

Reviewer #2 (Remarks to the Author):

The authors have addressed some of my criticisms, which improve the manuscript. However, I cannot find the experiments that they have listed in their response to the review. I quote "Since re-expression of WT p53 would arrest the cells and make cell cycle synchronization impossible, we performed mutp53 knockdown in C33A (Fig. 3D-E). Indeed, depletion of mutp53 rescued the inhibitory regulation of TopBP1/Treslin interaction in mid-late S phase (Fig. 3D-E), supporting a role for mutp53 in bridging

TopBP1/Treslin interaction in mid-late S phase."

The C33A knockdown experiments are in figure 7. I see no experiments where the mutant p53 is knocked down followed by a demonstration of the interaction between TopBP1/Treslin. Figure 3D-E are DNA fiber length measurements.

This needs to be corrected by the authors before the manuscript can be accepted.

Response:

We appreciate the reviewer's comment. We apologize for referring the experiment to the wrong figure number. The mutp53 knockdown in C33A cells is presented in Fig. 2D-E, instead of Fig. 3D-E. As shown in Fig. 2D-E (for your convenience, I attach Fig. 2D-E below), depletion of mutp53 rescued the inhibitory regulation of TopBP1/Treslin interaction in mid-late S phase, supporting a role for mutp53 in bridging TopBP1/Treslin interaction in mid-late S phase. The figure labeling in the manuscript is correct.

Fig. 2 in the manuscript